# DPP8/DPP9 inhibition elicits canonical Nlrp1b inflammasome hallmarks in murine macrophages

Nathalia M de Vasconcelos[1,2], Gwendolyn Vliegen[3], Amanda Gonçalves[2,4,5], Emilie De Hert[3], Rosa Martín-Pérez[6], Nina Van Opdenbosch[1,2,6], Anvesh Jallapally[7], Ruth Geiss-Friedlander[8], Anne-Marie Lambeir[3], Koen Augustyns[7], Pieter Van Der Veken[7], Ingrid De Meester[3], Mohamed Lamkanfi[1,2,6]

Activating germline mutations in the human inflammasome sensor NLRP1 causes palmoplantar dyskeratosis and susceptibility to Mendelian autoinflammatory diseases. Recent studies have shown that the cytosolic serine dipeptidyl peptidases DPP8 and DPP9 suppress inflammasome activation upstream of NLRP1 and CARD8 in human keratinocytes and peripheral blood mononuclear cells. Moreover, pharmacological inhibition of DPP8/DPP9 protease activity was shown to induce pyroptosis in murine C57BL/6 macrophages without eliciting other inflammasome hallmark responses. Here, we show that DPP8/DPP9 inhibition in macrophages that express a *Bacillus anthracis* lethal toxin (LeTx)–sensitive *Nlrp1b* allele triggered significantly accelerated pyroptosis concomitant with caspase-1 maturation, ASC speck assembly, and secretion of mature IL-1β and IL-18. Genetic ablation of ASC prevented DPP8/DPP9 inhibition-induced caspase-1 maturation and partially hampered pyroptosis and inflammasome-dependent cytokine release, whereas deletion of caspase-1 or gasdermin D triggered apoptosis in the absence of IL-1β and IL-18 secretion. In conclusion, blockade of DPP8/DPP9 protease activity triggers rapid pyroptosis and canonical inflammasome hallmarks in primary macrophages that express a LeTx-responsive *Nlrp1b* allele.

## Introduction

Inflammasomes play critical roles in the host response against pathogenic invasion and in inflammatory responses elicited by a diversity of host-derived and environmental stressors (Lamkanfi & Dixit, 2012, 2014). Canonical inflammasomes are assembled when the germline-encoded intracellular pattern recognition receptors nucleotide-binding oligomerization domain, leucine rich repeat and pyrin domain containing (NLRP)1, NLRP3, NLRC4, absent in melanoma (AIM2), and Pyrin detect so-called pathogen-associated molecular patterns and damage-associated molecular patterns. These cytosolic multi-protein platforms promote proximity-induced auto-activation of the cysteine protease caspase-1, which in turn cleaves the cytosolic precursor forms of interleukin (IL)-1β and IL-18 into mature inflammatory cytokines. Furthermore, inflammatory caspase-driven cleavage of gasdermin D (GSDMD) causes oligomerization and membrane internalization of the amino-terminal pore-forming domain, resulting in the hallmark features of pyroptosis including plasma membrane perforation, cell lysis, and the extracellular release of the soluble intracellular content (Kayagaki et al, 2015; Shi et al, 2015; Aglietti et al, 2016; Ding et al, 2016; Liu et al, 2016; Sborgi et al, 2016). Whereas canonical inflammasomes drive caspase-1 activation directly, intracellular Gram-negative lipopolysaccharides activate the inflammatory caspases -4, -5, and -11 in a signalling cascade referred to as the non-canonical inflammasome pathway (Kayagaki et al, 2011). The latter inflammatory caspases elicit pyroptosis autonomously, while engaging the NLRP3 inflammasome downstream of GSDMD for caspase-1–mediated secretion of IL-1β and IL-18 (Kayagaki et al, 2011).

Gain-of-function mutations in the inflammasome pattern recognition receptors NLRP3, Pyrin, and NLRC4 cause systemic autoinflammatory diseases that are frequently characterized by periodic fevers that last 3 to 7 d and recur every 2 to 12 wk, along with skin rash, abdominal pain, and other symptoms (Van Gorp et al, 2017). Notably, gain-of-function mutations in NLRP1 cause early onset skin inflammatory and epithelial dyskeratosis syndromes, whereas recurring fever is more variable in patients with *NLRP1* mutations (Grandemange et al, 2016; Zhong et al, 2016). In addition to the prototypical PYD, NACHT, and Leucine-rich repeat domains found in other NLRP family members, NLRP1 contains a unique carboxy-terminus extension that harbours a function-to-find (FIIND) domain and a CARD. The FIIND domain is an autoproteolytic domain that is uniquely shared between NLRP1 and CARD8 and undergoes

[1]Department of Internal Medicine, Ghent University, Ghent, Belgium   [2]VIB-UGhent Center for Inflammation Research, VIB, Ghent, Belgium   [3]Laboratory of Medical Biochemistry, Department of Pharmaceutical Sciences, University of Antwerp, Antwerp, Belgium   [4]VIB Bioimaging Core, VIB, Ghent, Belgium   [5]Department of Biomedical Molecular Biology, Ghent University, Ghent, Belgium   [6]Janssen Immunosciences, World Without Disease Accelerator, Pharmaceutical Companies of Johnson & Johnson, Beerse, Belgium   [7]Laboratory of Medicinal Chemistry, Department of Pharmaceutical Sciences, University of Antwerp, Antwerp, Belgium   [8]Institut für Molekularbiologie, Universitätsmedizin Göttingen, Göttingen, Germany

Correspondence: mlamkanf@its.jnj.com

posttranslational autocleavage as a prerequisite for ligand–induced activation (D'Osualdo et al, 2011; Finger et al, 2012; Frew et al, 2012).

Rodents lack a CARD8 homolog, but encode three orthologous *NLRP1* genes: *Nlrp1a*, *Nlrp1b*, and *Nlrp1c* (Boyden & Dietrich, 2006). Murine *Nlrp1c* is considered a pseudogene, whereas both Nlrp1a and Nlrp1b are established inflammasome sensors. An N-ethyl-N-nitrosourea mutagenesis screen for dominant mutations identified an activating *Nlrp1a*[Q593P] mutation that drove IL-1–dependent leukopenia in unchallenged mice by promoting excessive inflammasome activation and pyroptosis in hematopoietic progenitor cells (Masters et al, 2012). *Bacillus anthracis* lethal toxin (LeTx) is a well-defined biochemical virulence factor that potently triggers activation of the Nlrp1b inflammasome and pyroptosis in macrophages of genetically susceptible inbred mouse and rat strains (Boyden & Dietrich, 2006; Moayeri et al, 2010). Murine *Nlrp1b* is highly polymorphic, encoding five different alleles that drive macrophage susceptibility to LeTx in various inbred mouse strains (Boyden & Dietrich, 2006). Allele 1—found in 129S and BALB/c mice—and allele 5—found in the CAST/EiJ mouse strain—promote macrophage susceptibility to LeTx intoxication. On the other hand, allele 2 of A/J and C57BL/6J (B6) mice and alleles 3 and 4 in other inbred strains do not respond to LeTx and confer resistance to LeTx-induced pyroptosis in macrophages from these inbred strains (Boyden & Dietrich, 2006; Moayeri et al, 2010). Although it cannot be ruled out that the LeTx-unresponsive B6-derived *Nlrp1b* allele may have yet undiscovered activities, no studies to date have formally established that it is capable of eliciting inflammasome activation in response to endogenous, environmental, microbial, and pharmacological agents, and inflammasome activation upon LeTx intoxication has only been formally demonstrated in the presence of allele 1 of *Nlrp1b* (Boyden & Dietrich, 2006; Van Opdenbosch et al, 2014).

Recent studies have shown that pharmacological inhibitors of the S9B family of post-proline dipeptidyl peptidases (DPP)8 and DPP9 activate NLRP1 and CARD8 to induce pyroptosis in human keratinocytes, the human monocytic-like cell line THP-1, and in primary peripheral blood mononuclear cells, respectively (Johnson et al, 2018; Zhong et al, 2018). A recent report showed that DPP8/DPP9 inhibition in human keratinocytes elicited the known hallmark features of canonical inflammasome activation, including caspase-1 autocleavage, apoptosis-associated speck-like protein containing a CARD (ASC) speck formation, and secretion of mature IL-1β and IL-18 (Zhong et al, 2018). However, it is less clear whether these inflammasome responses are elicited upon DPP8/DPP9 inhibition in human and murine mononuclear cells (Okondo et al, 2017; Johnson et al, 2018; Zhong et al, 2018). The competitive pan-DPP inhibitor Val-*boro*Pro (VBP) (Coutts et al, 1996) and the more selective DPP8/DPP9 protease inhibitor 1G244 (Jiaang et al, 2005) were reported to induce pyroptosis without eliciting ASC speck assembly, ASC-dependent caspase-1 auto-maturation, or maturation and secretion of IL-1β from murine B6 macrophages, and the immortalized monocytic cell line RAW 264.7 (Okondo et al, 2017, 2018). However, B6 macrophages lack a LeTx-responsive *Nlrp1b* allele (Boyden & Dietrich, 2006; Van Opdenbosch et al, 2014), and the BALB/c-derived RAW 264.7 cell line lacks expression of the inflammasome adaptor protein ASC (Pelegrin et al, 2008). These observations prompted us to evaluate the inflammasome responses that are elicited by inhibition of DPP8/DPP9 in primary BMDM of mice that either express or lack LeTx-responsive *Nlrp1b* alleles. Our results demonstrate that DPP8/DPP9 inhibition significantly sensitizes macrophages that express a LeTx-responsive *Nlrp1b* allele to rapid pyroptosis induction concomitant with caspase-1 maturation, ASC speck assembly, and secretion of mature IL-1β and IL-18. Moreover, we show that genetic ablation of ASC in macrophages with a LeTx-responsive *Nlrp1b* allele prevented DPP8/DPP9 inhibition–induced caspase-1 maturation and partially hampered pyroptosis and inflammasome-dependent cytokine release, whereas deletion of caspase-1 or GSDMD triggered apoptosis in the absence of IL-1β and IL-18 secretion. In conclusion, our results demonstrate that DPP8/DPP9 protease activity suppresses caspase-1 maturation, ASC speck formation, cytokine secretion, and rapid pyroptosis in macrophages that express a LeTx-responsive *Nlrp1b* allele.

# Results

## Hemizygous expression of a LeTx-responsive *Nlrp1b* allele in C57BL/6J macrophages accelerates DPP8/DPP9 inhibition–induced pyroptosis

Wild-type B6 mice lack a LeTx-responsive *Nlrp1b* allele, rendering their BMDMs insensitive to *B. anthracis* LeTx-induced pyroptosis, but hemizygous expression of a LeTx-responsive *Nlrp1b* allele from a 129S1-derived BAC (B6[Nlrp1b+] mice) fully rescues LeTx-induced pyroptosis (Boyden & Dietrich, 2006; de Vasconcelos et al, 2018; Van Opdenbosch et al, 2014). To examine whether a LeTx-responsive *Nlrp1b* allele impacts cell death induction by the pan-DPP inhibitor VBP (Fig S1A), we monitored the induction of pyroptotic cell lysis over time in B6 and B6[Nlrp1b+] BMDMs in function of the number of nuclei stained with the membrane-impermeant DNA-intercalating dye Sytox Green (SG). Consistent with published reports (Okondo et al, 2017, 2018), by 6 h, VBP had induced low levels of pyroptosis (~20%) in B6 BMDMs, which gradually increased to reach a plateau at approximately 50% cytotoxicity by 24 h (Fig 1A). All tested VBP concentrations (10, 25, and 40 μM) induced cell death with similar kinetics and maximal cytotoxicity levels (Fig 1A), which was consistent with the observation that exogenous DPP9 activity was fully inhibited in lysates of BMDMs that had been pre-treated with 10 μM VBP (Fig S2A and B and Table S1). Notably, cell death induction was substantially faster in B6[Nlrp1b+] BMDMs at all tested concentrations with a 30% half-maximal level of pyroptosis reached within 3 h in B6[Nlrp1b+] BMDMs, whereas this same level of cytotoxicity required about 8 h in B6 macrophages (Fig 1A). Titration studies further showed that 1 μM VBP was sufficient to induce robust pyroptosis in B6[Nlrp1b+] BMDMs, whereas 0.1 μM of the inhibitor failed to trigger a cytotoxic response (Fig S2C).

Acetylation of VBP (aVBP, Fig S1B) prevents inhibition of DPP family members (Connolly et al, 2008). In agreement with inhibition of DPP protease activity being responsible for VBP's cytotoxic effect, aVBP failed to induce cytotoxicity in both B6 and B6[Nlrp1b+] BMDMs (Fig 1B). We next evaluated a cyclic VBP analog (cVBP, Fig S1) that was more than 100-fold attenuated in its ability to inhibit DPP9 activity in vitro (IC$_{50}$ cVBP = 1.70 ± 0.08 μM versus IC$_{50}$ VBP = 15.3 ± 0.8 nM) (Fig S2D and E and Table S1). All examined cVBP concentrations (10, 25, and 40 μM) failed to induce cell death in B6 BMDMs (Fig 1C).

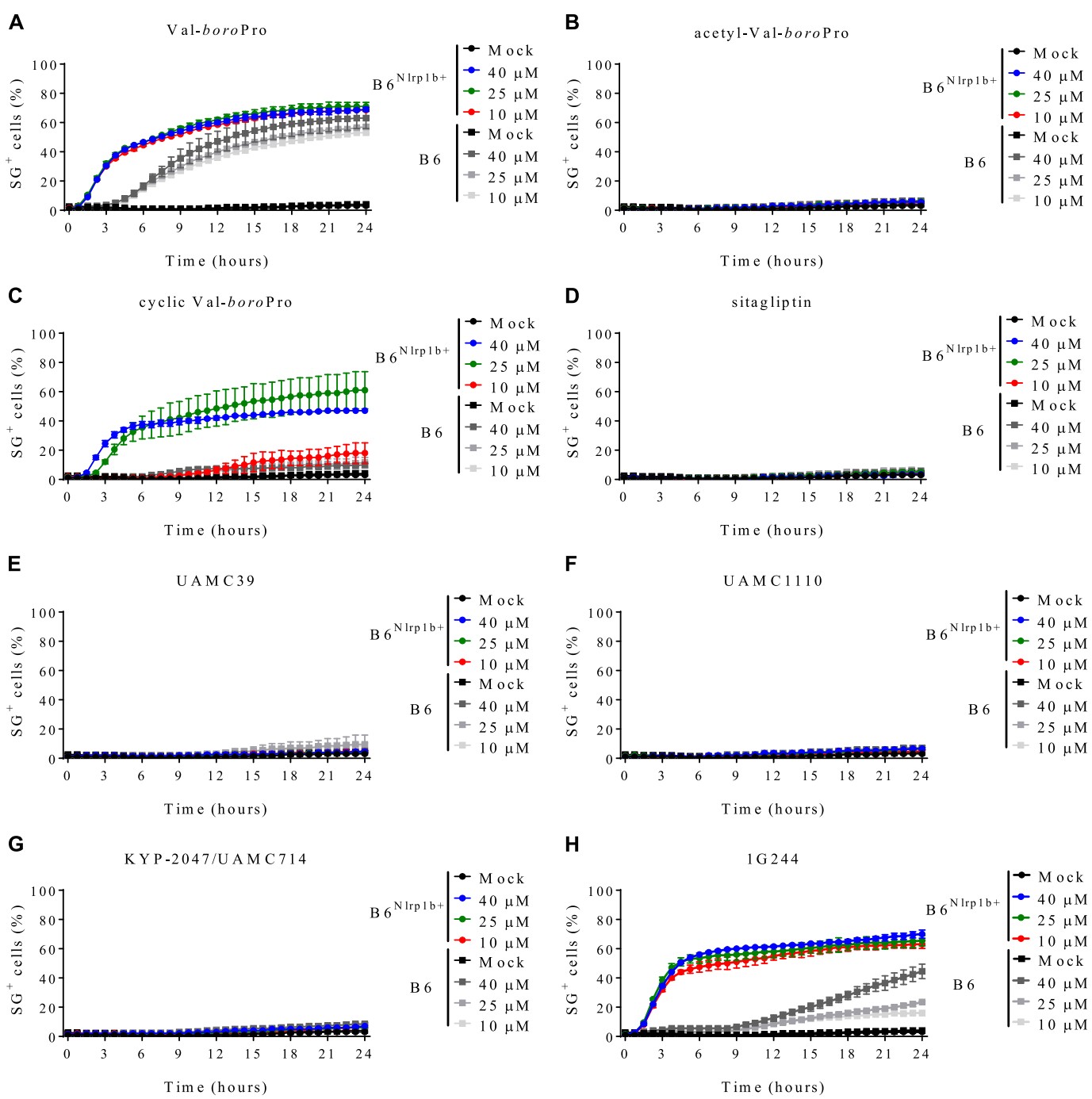

**Figure 1. Transgene expression of Nlrp1b sensitizes BMDMs to cell death upon DPP8/DPP9 inhibition.**
**(A–H)** B6 or B6^Nlrp1b+ BMDMs were left untreated or treated with 10, 25, or 40 μM of VBP (A), acetyl-VBP (B), cyclic-VBP (C), sitagliptin (D), UAMC39 (E), UAMC1110 (F), KYP-2047/UAMC714 (G), or 1G244 (H) in media containing SG and imaged on an Incucyte platform. In all graphs, the number of positive cells was quantified relative to a Triton X100–treated well considered 100%. Values represent mean ± SD of technical duplicates of a representative experiment from three biological repeats.

Contrastingly, however, cVBP induced significant pyroptosis in B6^Nlrp1b+ BMDMs (Fig 1C), although cell death induction by 10 μM cVBP still was attenuated and markedly delayed when compared with the cytotoxic response of VBP in B6^Nlrp1b+ BMDMs (Fig 1A and C).

VBP is a nonselective inhibitor of DPP8 and DPP9 that also inhibits the prolyl DPPs DPP4, fibroblast activation protein (FAP), and prolyl oligopeptidase (PREP) (Waumans et al, 2015). We evaluated

inhibitors with a more confined selectivity profile to better define the DPP family members that may contribute to the cytotoxic effect of VBP in macrophages. Sitagliptin, UAMC39, UAMC1110, and KYP-2047/UAMC714 (Fig S1D–G) specifically target DPP4, DPP2, FAP, and PREP, respectively (Jansen et al, 2014; Jarho et al, 2004; Kim et al, 2005; Senten et al, 2004). We first established that treatment of BMDMs with 10 μM sitagliptin, UAMC39, UAMC1110, or KYP-2047/

UAMC714 sufficed to reach intracellular concentrations of the respective inhibitors that maximally inhibit their protease targets in BMDMs (Fig S3 and Table S2). However, none of the listed inhibitors triggered a cell death response in B6 and B6^Nlrp1b+ BMDMs at concentrations of 10, 25, or 40 μM, respectively (Fig 1D–G). Overall, these results suggest that inhibition of DPP4, DPP2, FAP, and PREP does not reflect VBP's ability to induce pyroptosis, which may be selectively linked to its activity on DPP8 and DPP9. Compound 1G244 (Fig S1H) is highly selective for DPP8/DPP9 over other DPP family members, albeit targeting these enzymes with reduced potency compared with VBP (Wu et al, 2009). Cell lysates of BMDMs that have been treated with 10 μM 1G244 inhibited DPP8/DPP9 activity by 98% in our in vitro assay (Fig S4A and B and Table S3), suggesting that this concentration suffices to reach intracellular 1G244 concentrations that support near-complete target engagement. Consistently, B6^Nlrp1b+ BMDMs rapidly underwent pyroptosis when treated with 10 μM 1G244 with a 30% half-maximal level of pyroptosis reached within 2 h, and a plateau of about 60% pyroptosis evident by 6 h posttreatment (Fig 1H). Increasing the 1G244 treatment concentrations to 25 μM or 40 μM did not alter cell death kinetics in B6^Nlrp1b+ BMDMs (Fig 1H), in agreement with our in vitro observations demonstrating near full DPP8/DPP9 inhibition with 10 μM 1G244 on macrophages (Fig S4A and B and Table S3). Titration studies further showed that 1 μM 1G244 induced a reduced level of pyroptosis in B6^Nlrp1b+ BMDMs, whereas 0.1 μM of the inhibitor failed to trigger a cytotoxic response (Fig S4C). Consistent with the reported findings (Waumans et al, 2016; Okondo et al, 2017), 1G244 and VBP—at concentrations of 10 μM and 25 μM—elicited a swift and robust cytotoxic response in the BALB/c-derived monocytic cell line J774.A1 that natively expresses a LeTx-responsive Nlrp1b allele (Fig S5). As seen with VBP (Fig 1A), the cytotoxic response of B6 BMDMs that had been treated with 10 μM or 25 μM 1G244 was markedly slower and attenuated with pyroptosis commencing around 10 h posttreatment and reaching ~20% of the cell population by 24 h (Fig 1H). A concentration of 40 μM 1G244 reached a more potent 40% pyroptosis level by 24 h, although the cell death kinetics remained slow (Fig 1H). Collectively, these results establish that expression of a LeTx-responsive Nlrp1b allele drives a fast and prominent pyroptosis response upon DPP8/DPP9 inhibition in primary macrophages and in the BALB/c-derived monocytic cell line J774.A1.

### DPP8/DPP9 inhibition elicits caspase-1 cleavage and secretion of mature IL-1β and IL-18 from macrophages with a LeTx-responsive Nlrp1b inflammasome

LeTx induces pyroptosis in LPS-primed B6^Nlrp1b+ BMDMs concomitant with the induction of Nlrp1b-dependent caspase-1 auto-cleavage and maturation and secretion of the inflammatory cytokines IL-1β and IL-18 (Boyden & Dietrich, 2006; Van Opdenbosch et al, 2014, 2017). VBP and 1G244 were recently shown to induce pyroptosis in B6 macrophages without eliciting caspase-1 auto-maturation, or maturation and secretion of IL-1β in these cells (Okondo et al, 2017, 2018). We sought to examine how these different inflammasome parameters were impacted in VBP-, cVBP-, and 1G244-treated B6^Nlrp1b+ BMDMs. Macrophages were primed with LPS for 3 h before treatment with VBP, cVBP, 1G244, or aVBP at different concentrations (0, 10 and 40 μM) and different incubation times (4, 8, and 24 h). As expected (Boyden & Dietrich, 2006; Van Opdenbosch et al, 2014, 2017), LPS priming does not activate caspase-1 and B6^Nlrp1b+ BMDMs that had been primed with LPS for up to 24 h failed to induce caspase-1 maturation into its prototypic p20 subunit (Fig 2A). However, VBP triggered marked caspase-1 cleavage with the catalytic p20 subunit being evident in immunoblots of B6^Nlrp1b+ BMDMs that had been treated with both tested VBP concentrations (10 or 40 μM) and any of the examined timepoints (4, 8, and 24 h). Consistent with our pyroptosis titration studies (Fig S2C), 1 μM VBP was sufficient to trigger caspase-1 maturation, whereas 0.1 μM VBP failed to induce caspase-1 auto-cleavage (Fig S6A). The inactive VBP analog aVBP also failed to elicit caspase-1 maturation after 24 h (Fig 2A), demonstrating the specificity of this response. Consistent with the reduced DPP8/DPP9 inhibition by cVBP, caspase-1 p20 cleavage was detected only in lysates of macrophages that had been treated with the highest concentration (40 μM) (Fig 2B). Corroborating our other findings (Fig 1H), 1G244 efficiently promoted caspase-1 auto-maturation into the catalytic p20 subunit as evident from caspase-1 immunoblots of B6^Nlrp1b+ BMDMs that had been incubated with 1G244 for any of the examined time periods (4, 8, and 24 h) and concentrations (10 or 40 μM) (Fig 2C). Moreover, 10 μM 1G244 was most optimal to induce caspase-1 cleavage considering that 1 μM of the inhibitor triggered weak caspase-1 processing (Fig S6B). Together, these results establish that DPP8/DPP9 inhibition elicits efficient caspase-1 maturation concurrent with the induction of pyroptosis in macrophages expressing a LeTx-responsive Nlrp1b allele.

Although LPS priming increases cytosolic expression levels of proIL-1β, it is well-established that LPS priming is dispensable for LeTx-induced caspase-1 activation by the Nlrp1b inflammasome (Boyden & Dietrich, 2006; Van Opdenbosch et al, 2014, 2017). We treated naive and LPS-primed macrophages with VBP, 1G244, and cVBP to examine whether LPS priming impacts DPP8/DPP9 inhibition–induced caspase-1 auto-processing in macrophages that express the LeTx-responsive Nlrp1b allele. Comparable levels of caspase-1 cleavage were detected in immunoblots of naive and LPS-pretreated B6^Nlrp1b+ BMDMs that had subsequently been stimulated with VBP, 1G244, or cVBP (Fig 2D), adding further support to the notion that LPS-priming is dispensable for DPP8/DPP9 inhibition–induced caspase-1 maturation by the Nlrp1b inflammasome. As expected, cytosolic levels of proIL-1β were markedly up-regulated in LPS-primed B6^Nlrp1b+ BMDMs relative to proIL-1β expression in naive macrophages, whereas DPP8/DPP9 inhibition by VBP, 1G244, or cVBP was required to induce IL-1β maturation in LPS-primed macrophages (Fig 2D). Encouraged by these observations, we next measured the levels of secreted IL-1β and IL-18 in response to DPP8/DPP9 inhibition during 4, 8, or 24 h. As expected, B6^Nlrp1b+ BMDMs that were primed with LPS for 4, 8, or 24 h did not secrete IL-1β or IL-18 in their extracellular medium (Fig 2E and F). However, LPS-primed B6^Nlrp1b+ BMDMs potently released IL-1β (Figs 2E and S7A) and IL-18 (Fig 2F) as early as 4 h post-VBP treatment. Consistent with the reduced DPP8/DPP9 inhibition effect of cVBP and its attenuated ability to trigger pyroptosis, this compound was less efficient in eliciting extracellular secretion of IL-1β and IL-18 at 10 μM, although it triggered substantial extracellular release of IL-1β and IL-18 at 40 μM (Fig 2E and F). Moreover, the selective

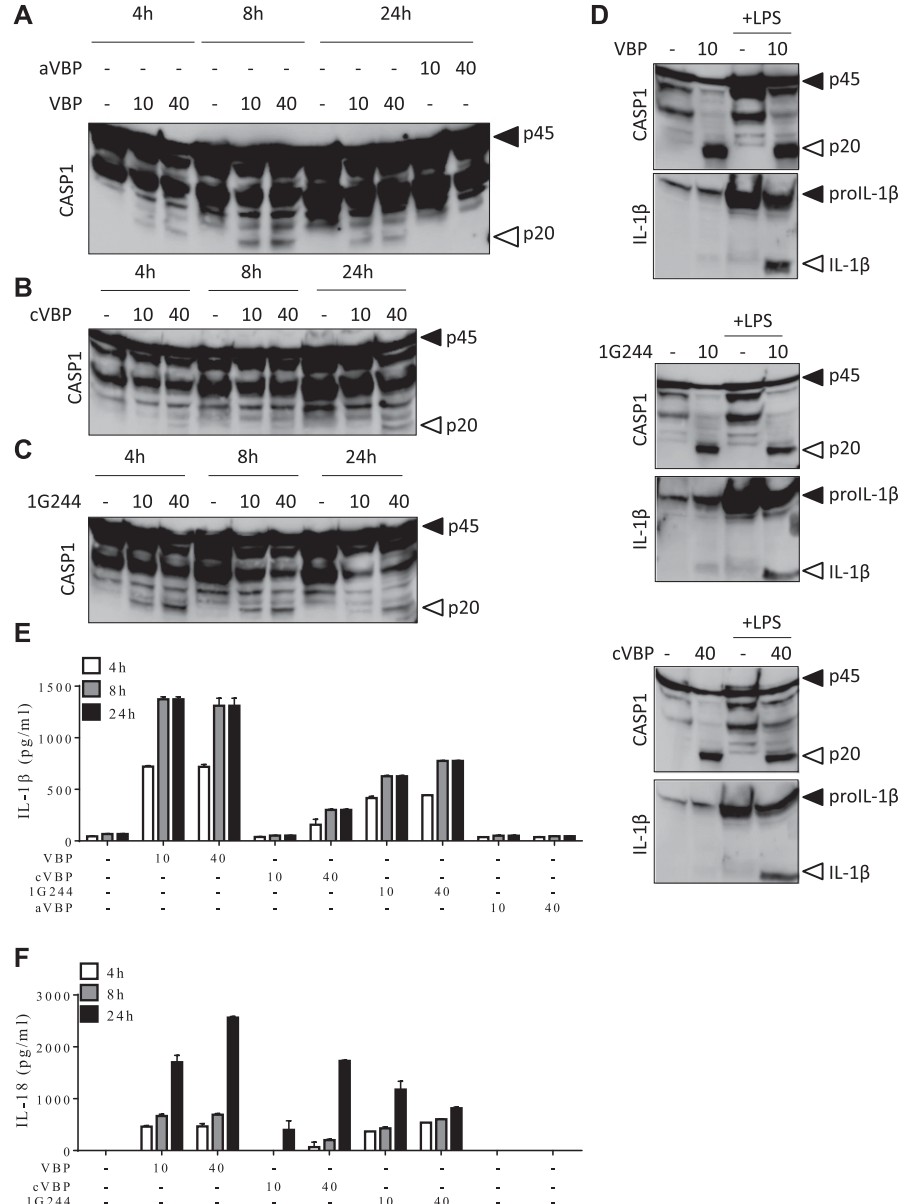

**Figure 2. Inhibition of DPP8/DPP9 activates the Nlrp1b inflammasome.**
**(A–C, E, F)** B6^Nlrp1b+ BMDMs were initially primed with LPS for 3 h, then treated with 10 μM or 40 μM of VBP, aVBP, cVBP, or 1G244 and collected after 4, 8, and 24 h. Combined cell lysate and supernatant samples were immunoblotted for the indicated proteins (A–C), and supernatants were analyzed for IL-1β (E) and IL-18 (F) secretion. (D) Combined cell lysate and supernatant samples of B6^Nlrp1b+ BMDMs either primed or not with LPS for 3 h and treated with 10 μM of VBP or 10 μM of 1G244 or 40 μM of cVBP were collected after 8 h and immunoblotted for the indicated proteins. Cytokine values represent mean ± SD of technical duplicates from three biological repeats. All data are representative from three biological repeats.

DPP8/DPP9 inhibitor 1G244 potently triggered secretion of IL-1β and IL-18 at each of the examined time points and drug concentrations, whereas the inactive VBP analog aVBP failed to induce cytokine secretion (Figs 2E and F and S7B). Collectively, these results demonstrate that DPP8/DPP9 inhibition–induced activation of the Nlrp1b inflammasome drives caspase-1 auto-processing into the catalytic p20 subunit and maturation and extracellular release of the inflammasome-dependent cytokines IL-1β and IL-18 concomitant with the induction of pyroptosis.

### Role of ASC in DPP8/DPP9 inhibition–mediated Nlrp1b inflammasome signaling

The bipartite inflammasome adaptor protein ASC oligomerizes in response to inflammasome-activating cues to form ASC specks (Lamkanfi & Dixit, 2012, 2014). These higher order structures frequently appear as a large (1–2 μm sized) perinuclear puncta and serve as scaffolds for caspase-1 and caspase-8 activation in inflammasome-activated cells (Gurung et al, 2014; Van Opdenbosch et al, 2014, 2017). Notably, DPP inhibitors VBP and 1G244 were shown to induce pyroptosis in the ASC-deficient immortalized monocytic cell line RAW 264.7 and in the ASC-deficient clones of the human monocytic cell line THP1 (Okondo et al, 2017). However, whether DPP8/DPP9 inhibition elicits ASC speck formation in ASC-sufficient macrophages and monocytes is not known. To examine the presence of ASC specks in B6^Nlrp1b+ macrophages, BMDMs were stimulated with 10 μM VBP or 1G244 for 4 or 8 h before cells were fixed and immunolabelled for ASC. Confocal image analysis of DAPI-counterstained cells showed marked ASC speck formation in response to VBP and 1G244 (Fig 3A). As a control, mock-treated cells

failed to display ASC specks (Fig 3A). Automated quantification of ASC specks in randomly chosen confocal micrographs confirmed the increase in ASC speck counts following DPP8/DPP9 inhibition (Fig 3B).

Having established that DPP8/DPP9 inhibition–induced inflammasome activation is accompanied by the formation of ASC specks, we bred ASC-deficient mice to B6$^{Nlrp1b+}$ animals to study the functional role of ASC in VBP- and 1G244-induced pyroptosis, caspase-1 auto-maturation, and IL-1$\beta$ secretion in Nlrp1b-sufficient macrophages. We (Van Opdenbosch et al, 2014) and others (Guey et al, 2014) previously reported that LeTx-induced pyroptosis continued unabated in ASC$^{-/-}$ macrophages, whereas caspase-1 maturation was abolished and secretion of IL-1$\beta$ attenuated. Notably, the maximal levels of pyroptosis induced by VBP and 1G244 were reduced by about 50% in B6$^{Nlrp1b+}$ASC$^{-/-}$ BMDMs compared with ASC-sufficient B6$^{Nlrp1b+}$ macrophages although cell death kinetics remained largely unaffected (Fig 3C and D). As expected (Guey et al, 2014; Van Opdenbosch et al, 2014), we observed comparable kinetics and maximal levels of pyroptosis in LeTx-treated B6$^{Nlrp1b+}$ASC$^{-/-}$ BMDMs and B6$^{Nlrp1b+}$ macrophages (Fig S8). Akin to reported findings in the monocytic cell line RAW 264.7 (Okondo et al, 2017, 2018) (that is naturally deficient in ASC expression [Pelegrin et al, 2008]), VBP- and 1G244-induced caspase-1 maturation was blunted in B6$^{Nlrp1b+}$ASC$^{-/-}$ BMDMs (Fig 3E and F). Moreover, DPP8/DPP9 inhibition–induced secretion of IL-1$\beta$ in culture supernatants of B6$^{Nlrp1b+}$ASC$^{-/-}$ BMDMs was reduced compared with those of B6$^{Nlrp1b+}$ cells (Fig 3G). Based on these findings, we conclude that ASC is required for DPP8/DPP9 inhibition–induced caspase-1 maturation while partially contributing to induction of pyroptosis and IL-1$\beta$ secretion by the Nlrp1b inflammasome.

### DPP8/DPP9 inhibition induces ASC-mediated apoptosis in caspase-1–deficient B6$^{Nlrp1b+}$ macrophages

It was recently shown that LeTx-intoxicated B6$^{Nlrp1b+}$ macrophages switch to an ASC- and caspase-8–dependent apoptotic program when failing to activate caspase-1 (Van Opdenbosch et al, 2017). We bred B6$^{Nlrp1b+}$ mice to caspase-1/11–deficient animals to analyse whether DPP8/DPP9 inhibition similarly triggers apoptosis in caspase-1–deficient macrophages. As expected, B6$^{Nlrp1b+}$ macrophages displayed a prototypical "necrotic ballooning" phenotype that was indicative of pyroptosis induction following DPP8/DPP9 inhibition by VBP or 1G244 (Fig 4A). Contrastingly, DPP8/DPP9 inhibition in B6$^{Nlrp1b+}$C1$^{-/-}$C11$^{-/-}$ macrophages induced apoptosis as evidenced by the appearance of apoptotic bodies (Fig 4A). We further bred B6$^{Nlrp1b+}$C1$^{-/-}$C11$^{-/-}$ mice to ASC$^{-/-}$ mice to investigate whether DPP8/DPP9 inhibition–induced apoptosis was mediated by ASC specks. Notably, VBP- and 1G244-treated B6$^{Nlrp1b+}$C1$^{-/-}$C11$^{-/-}$ASC$^{-/-}$ macrophages remained alive and were fully rescued from apoptosis and pyroptosis induction (Fig 4A), akin to reported results for LeTx-stimulated B6$^{Nlrp1b+}$C1$^{-/-}$C11$^{-/-}$ASC$^{-/-}$ macrophages (Van Opdenbosch et al, 2017). As expected, cell lysis–associated internalization of the cell-impermeant DNA-intercalating agent SG occurred in pyroptotic 1G244-treated B6$^{Nlrp1b+}$ macrophages specifically, but not in early apoptotic B6$^{Nlrp1b+}$C1$^{-/-}$C11$^{-/-}$ macrophages or cell death–resistant B6$^{Nlrp1b+}$C1$^{-/-}$C11$^{-/-}$ASC$^{-/-}$ macrophages (Fig 4B). Similar results were

obtained in VBP-treated macrophages (Fig 4C). Moreover, VBP-induced caspase-1 maturation in B6$^{Nlrp1b+}$ macrophages, whereas marked ASC-dependent caspase-8 cleavage was observed in cell lysates of B6$^{Nlrp1b+}$C1$^{-/-}$C11$^{-/-}$ macrophages (Fig 4D). Finally, culture supernatants of LPS-primed B6$^{Nlrp1b+}$ macrophages that were subsequently treated with VBP or 1G244 contained substantial amounts of IL-1$\beta$ (Fig 4E) and IL-18 (Fig 4F), whereas supernatants of B6$^{Nlrp1b+}$C1$^{-/-}$C11$^{-/-}$ and B6$^{Nlrp1b+}$C1$^{-/-}$C11$^{-/-}$ASC$^{-/-}$ cells lacked these cytokines (Fig 4E and F). These results demonstrate that inhibition of DPP8/DPP9 protease activity in caspase-1–deficient macrophages induces ASC-dependent caspase-8 activation and apoptosis without secretion of inflammasome-dependent cytokines.

### Apoptosis induction in GSDMD-deficient macrophages

DPP8/DPP9 inhibition was previously shown to trigger apoptosis in GSDMD-deficient clones of the human leukemic cell line THP-1 and the ASC-deficient murine monocytic cell line RAW264.7 (Taabazuing et al, 2017), akin to published findings in LeTx-stimulated GSDMD$^{-/-}$ macrophages (de Vasconcelos et al, 2018). We generated B6$^{Nlrp1b+}$GSDMD$^{-/-}$ macrophages to address whether DPP8/DPP9 inhibition in primary macrophages that lack GSDMD expression similarly undergo apoptosis. The appearance of apoptotic bodies and other hallmark features of apoptosis were evident in confocal micrographs of B6$^{Nlrp1b+}$GSDMD$^{-/-}$ macrophages that were treated for 4 h with DPP8/DPP9 inhibitors VPB and 1G244 (Fig 5A). As controls, GSDMD-sufficient B6$^{Nlrp1b+}$ macrophages showed a pyroptotic morphology under these conditions, whereas mock-treated controls of either genotype had a healthy appearance (Fig 5A). Remarkably, unlike in apoptotic B6$^{Nlrp1b+}$C1$^{-/-}$C11$^{-/-}$ macrophages (Fig 4B), a kinetic analysis of plasma membrane rupture in apoptotic B6$^{Nlrp1b+}$GSDMD$^{-/-}$ macrophages showed that cell lysis was only slightly delayed compared with pyroptotic B6$^{Nlrp1b+}$ macrophages (Fig 5B and C), indicating that late apoptotic GSDMD-deficient cells swiftly proceeded into secondary necrosis. However, culture supernatants of LPS-primed B6$^{Nlrp1b+}$GSDMD$^{-/-}$ macrophages contained markedly reduced levels of IL-1$\beta$ and IL-18 (Fig 5D and E), suggesting that inflammasome-induced cytokine secretion was primarily GSDMD-driven. In conclusion, these results demonstrate that DPP8/DPP9 inhibitors VBP and 1G244 potently induce apoptosis in primary macrophages lacking GSDMD expression.

# Discussion

DPP8 and DPP9 are members of the DPP S9B family that cleave peptide substrates after a prolyl bond (Waumans et al, 2015). In addition to DPP8 and DPP9, this family comprises the serine proteases DPP4, FAP, and PREP (Waumans et al, 2015). Myeloid cells predominantly express DPP8 and DPP9 (Matheeussen et al, 2013). Early studies recognized the pro-inflammatory properties of VBP, a broad-spectrum small molecule inhibitor of DPP protease activity (Adams et al, 2004; Nemunaitis et al, 2006). More recently, VBP was shown to be unable to induce elevation of serum G-CSF and CXCL1/KC levels in mice lacking caspases -1 and -11 (Okondo et al, 2017), highlighting that inflammasome activation plays a central role in

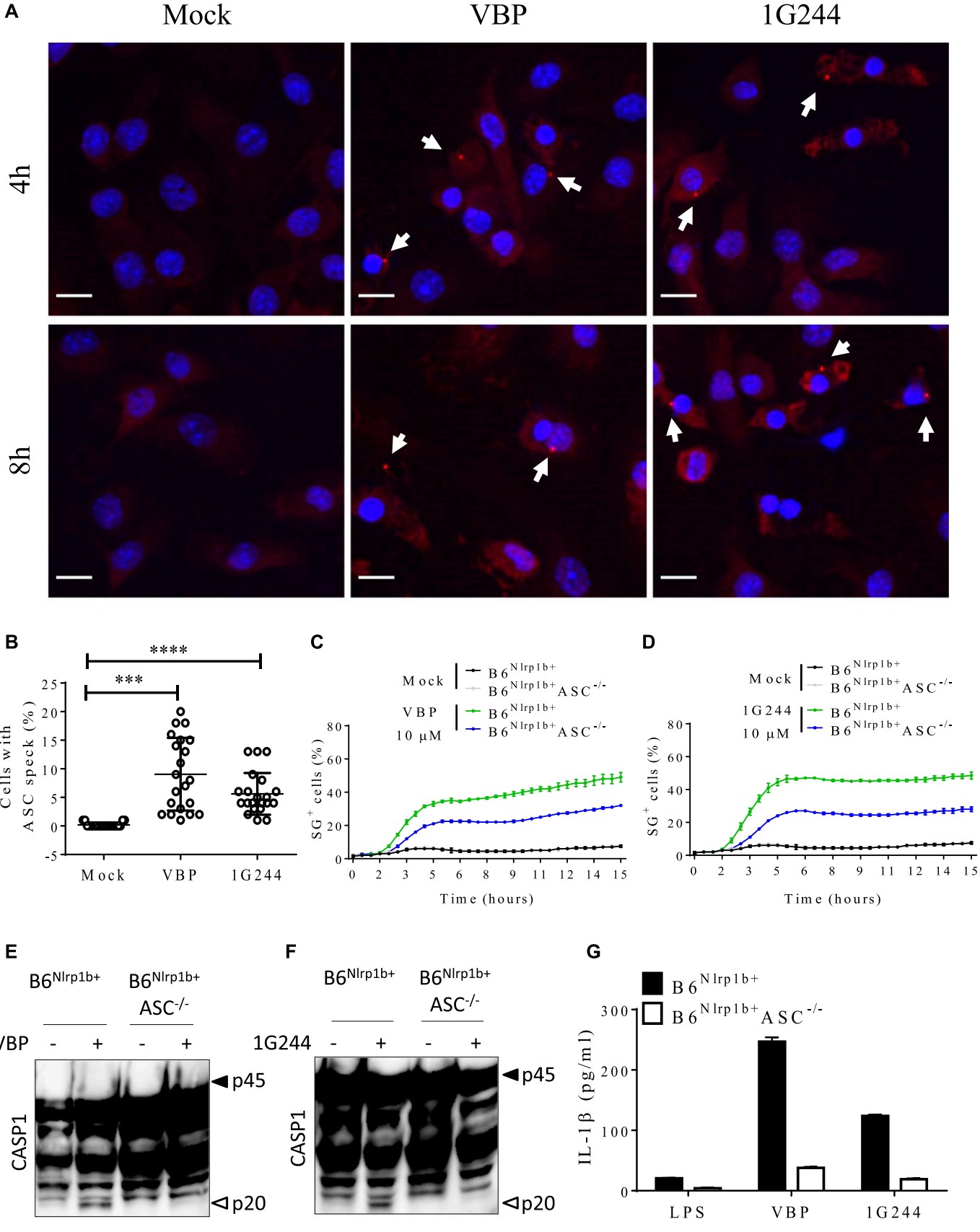

VBP-induced immune stimulation in vivo. Consistently, VBP and the more selective DPP8/DPP9 inhibitor 1G244 were shown to engage the inflammasome adaptor protein CARD8 to induce pyroptosis of a panel of acute myeloid leukemia (AML) cell lines and primary AML samples (Johnson et al, 2018). In addition, DPP8/DPP9 protease activity and physical binding between DPP9 and the FIIND domain of human NLRP1 were shown to repress activation of the NLRP1 inflammasome in human monocytes and primary keratinocytes (Zhong et al, 2018). Notably, DPP8/DPP9 inhibition in human monocytes and primary keratinocytes induced caspase-1 auto-cleavage, ASC speck formation, and secretion of mature IL-1β and IL-18 (Zhong et al, 2018), whereas VBP and 1G244 were reported to induce pyroptosis without eliciting ASC speck assembly, ASC-dependent caspase-1 auto-maturation, or maturation and secretion of IL-1β from human THP-1 cells, murine B6 macrophages, and the immortalized monocytic cell line RAW 264.7 (Okondo et al, 2017, 2018; Johnson et al, 2018). However, a comprehensive analysis of DPP8/DPP9-repressed inflammasome activation in primary macrophages expressing an *Nlrp1b* allele that is responsive to LeTx has not been reported to date.

*Nlrp1b* allele 2 of B6 macrophages has not been decisively shown to support inflammasome assembly. Conversely, C57BL/6J macrophages express a functional *Nlrp1a* allele as highlighted by spontaneous induction of pyroptosis of hematopoietic progenitor cells and cytopenia in mice expressing an N-ethyl-N-nitrosourea–induced gain-of-function $Nlrp1a^{Q593P}$ mutation (Masters et al, 2012). Consistent with the notion that transgenic introduction of a 129S-derived *Nlrp1b* allele renders B6 mouse BMDMs susceptible to LeTx-induced pyroptosis (Boyden & Dietrich, 2006; Van Opdenbosch et al, 2014, 2017; de Vasconcelos et al, 2018), we showed here that a LeTx-responsive *Nlrp1b* allele markedly accelerated pyroptosis of B6 macrophages triggered by DPP8/DPP9 inhibition when compared with wild-type B6 macrophages that natively express a LeTx-resistant *Nlrp1b* allele. Consistent with the requirement of a responsive *Nlrp1b* allele for efficient pyroptosis induction by DPP8/DPP9 inhibitors, 1G244 more potently induced cytotoxicity in the BALB/c-derived myeloid leukemia cell line J774.A1 than in the wild-type B6 primary macrophages (Waumans et al, 2016). We hypothesize that the slower pyroptotic response seen in B6 macrophages may be mediated by the Nlrp1a inflammasome. However, addressing the role of the Nlrp1a inflammasome in DPP8/DPP9 inhibition–induced pyroptosis awaits the generation of B6 mice with a targeted *Nlrp1a* null allele. Regardless, we further established that DPP8/DPP9 inhibition–induced activation of the Nlrp1b inflammasome drives ASC speck assembly, ASC-mediated caspase-1 auto-processing into the catalytic p20 subunit, and maturation and extracellular release of the inflammasome-dependent

cytokines IL-1β and IL-18 concomitant with induction of pyroptosis. Our observation that ASC-deficiency in primary B6^Nlrp1b+^ASC^−/−^ BMDMs blunted caspase-1 automaturation, ASC speck assembly and maturation and secretion of inflammasome-dependent cytokines IL-1β and IL-18 is consistent with the reported absence of caspase-1 cleavage and defective IL-1β secretion from the VBP-treated RAW 264.7 cell line, which is naturally deficient in ASC expression (Okondo et al, 2017, 2018; Johnson et al, 2018). Moreover, we showed that ASC is critical for caspase-8 activation and apoptosis induction in VBP- and 1G244-treated B6^Nlrp1b+^ macrophages with a deficiency in caspases -1 and -11, akin to what was reported for LeTx-stimulated B6^Nlrp1b+^C1^−/−^C11^−/−^ macrophages (Van Opdenbosch et al, 2017). Finally, GSDMD-deficient B6^Nlrp1b+^macrophages also switched to an apoptotic program in response to DPP8/DPP9 inhibition, paralleling recent findings that B6^Nlrp1b+^GSDMD^−/−^ macrophages switch to Nlrp1b-dependent apoptosis when intoxicated with LeTx (de Vasconcelos et al, 2018). Future work should investigate why DPP8/DPP9 inhibitor–induced apoptosis progresses significantly faster to a lytic phase (as marked by SG incorporation) in B6^Nlrp1b+^GSDMD^−/−^ macrophages compared with apoptotic B6^Nlrp1b+^C1^−/−^C11^−/−^ BMDMs.

In conclusion, the presented work established that a LeTx-responsive Nlrp1b inflammasome markedly accelerates DPP8/DPP9 inhibitor–induced pyroptosis in primary macrophages and demonstrates that this is accompanied by all hallmark features of canonical inflammasome activation, including significant ASC speck assembly, ASC-dependent caspase-1 autocleavage, and maturation and secretion of inflammasome-dependent cytokines in the extracellular environment. Future studies should clarify under what conditions canonical inflammasome hallmarks accompany pyroptosis induction by DPP8/DPP9 inhibitors in primary human monocytes, macrophages, and keratinocytes of healthy individuals and diseased patients (Okondo et al, 2017, 2018; Taabazuing et al, 2017; Johnson et al, 2018; Zhong et al, 2018). Addressing this and other questions will shed additional light on NLRP1- and CARD8-mediated inflammasome activation, and aid translational work to target DPP8/DPP9 protease activity in AML and NLRP1-mediated skin dyskeratosis syndromes.

# Materials and Methods

### Mice

B6^Nlrp1b+^, ASC^−/−^, C1^−/−^11^−/−^, and GSDMD^−/−^ mice have been reported (Kuida et al, 1995; Mariathasan et al, 2004; Boyden & Dietrich, 2006; Kayagaki et al, 2015) and mice were interbred to

**Figure 3. Inhibition of DPP8/DPP9 elicits the inflammasome partially relying on ASC.**
(A, B) ASC specks were imaged from B6^Nlrp1b+^ BMDMs treated with 10 μM VBP or 1G244 for 4 h or 8 h. (A) Representative confocal micrographs from three different experiments depicting DAPI (blue) and ASC (red). Arrows indicate ASC specks. (B) Quantification of the number of cells containing an ASC speck after 8 h of treatment with inhibitors are plotted as the mean ± SD obtained in each mosaic of three different experiments. (C, D) B6^Nlrp1b+^ or B6^Nlrp1b+^ASC^−/−^ BMDMs were mock-treated or received 10 μM VBP (C) or 1G244 (D) in media containing SG and imaged on an IncuCyte platform. The number of positive cells was quantified relative to a Triton X100–treated well considered 100%. Values represent mean ± SD of technical duplicates of a representative experiment from three biological repeats. (F, G) B6^Nlrp1b+^ or B6^Nlrp1b+^ASC^−/−^BMDMs were treated with 10 μM of VBP (E, G) or 1G244 (F, G) for 8 h. Combined cell lysate and supernatant samples were immunoblotted for caspase-1 (E, F) and supernatants were probed for IL-1β (G) levels. Cytokine values represent mean ± SD of technical duplicates from three biological repeats. All data are representative from three biological repeats. Scale bars, 10 μm. ***P < 0.001; ****P < 0.0001 tested by one-way Anova with Tukey's multiple comparisons.

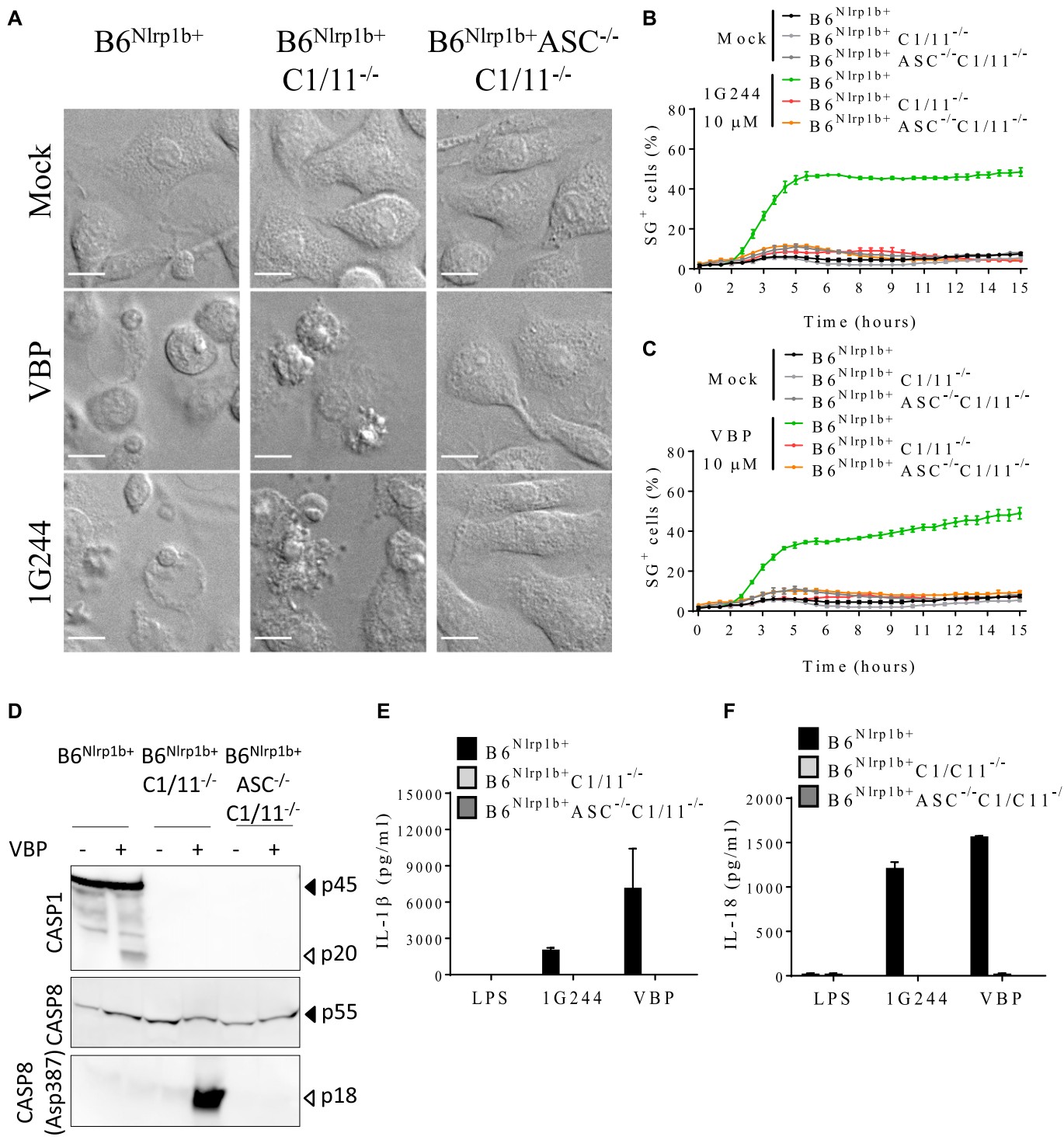

**Figure 4. DPP8/DPP9 inhibition elicits cell death and cytokine release dependent on caspase-1/-11.**
**(A)** B6[Nlrp1b+], B6[Nlrp1b+]C1/11[−/−], or B6[Nlrp1b+]C1/11[−/−]ASC[−/−] BMDMs were mock-treated or received 10 $\mu$M of VBP or 1G244 and imaged under a confocal microscope after 4 h. All scale bars, 10 $\mu$m. **(B, C)** B6[Nlrp1b+], B6[Nlrp1b+]C1/11[−/−], or B6[Nlrp1b+]C1/11[−/−]ASC[−/−] BMDMs were mock-treated or received 10 $\mu$M of 1G244 (B) or VBP (C) in media containing SG and imaged on an Incucyte platform. The number of positive cells was quantified relative to a Triton X100–treated well considered 100%. Values represent mean ± SD of technical duplicates of a representative experiment from three biological repeats. **(D)** B6[Nlrp1b+], B6[Nlrp1b+]C1/11[−/−], or B6[Nlrp1b+]C1/11[−/−]ASC[−/−] BMDMs were treated with 10 $\mu$M of VBP for 4 h and immunoblotted for the indicated proteins. **(E, F)** Supernatants from B6[Nlrp1b+], B6[Nlrp1b+]C1/11[−/−], or B6[Nlrp1b+]C1/11[−/−]ASC[−/−] BMDMs primed with LPS for 3 h and treated with 10 $\mu$M 1G244 or VBP for 8 h were assayed for IL-1$\beta$ (E) or IL-18 (F) levels. Cytokine values represent mean ± SD of technical duplicates from three biological repeats. All data are representative from three biological repeats.

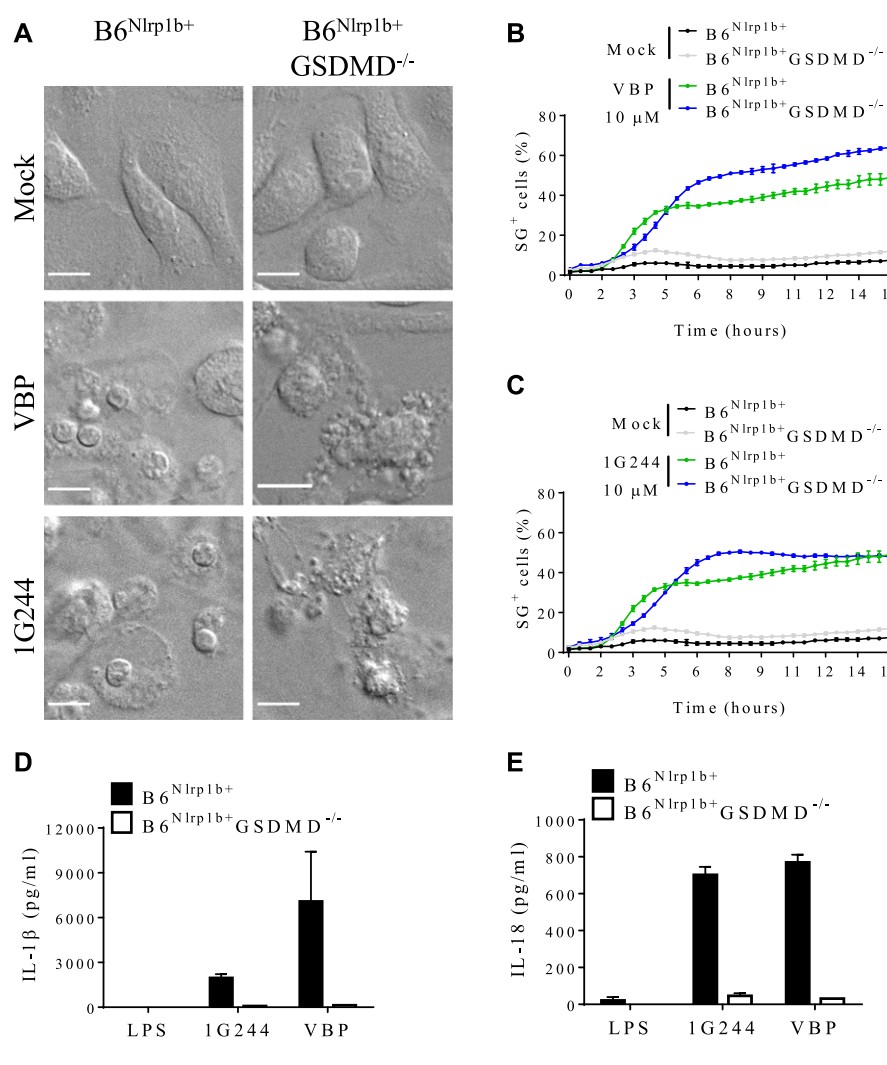

**Figure 5. Cell death and cytokine release upon inhibition of DPP8/DPP9 relies on GSDMD.**
**(A)** B6[Nlrp1b+] or B6[Nlrp1b+]GSDMD[−/−] BMDMs were either mock-treated or received 10 μM of VBP or 1G244 and imaged under a confocal microscope after 4 h. All scale bars, 10 μm. **(B, C)** B6[Nlrp1b+] or B6[Nlrp1b+]GSDMD[−/−] BMDMs were mock-treated or received 10 μM of VBP (B) or 1G244 (C) in media containing SG and imaged on an Incucyte platform. The number of positive cells was quantified relative to a Triton X100–treated well considered 100%. Values represent mean ± SD of technical duplicates of a representative experiment from three biological repeats. **(D, E)** Supernatants from B6[Nlrp1b+] or B6[Nlrp1b+] GSDMD[−/−] BMDMs primed with LPS for 3 h and treated with 10 μM VBP or 1G244 for 4 h were assayed for IL-1β (D) or IL-18 (E) levels. Cytokine values represent mean ± SD of technical duplicates from three biological repeats. All data are representative from three biological repeats.

obtain the desired genotypes on an hemizygous B6[Nlrp1b+] genetic background. B6 mice were originally purchased from Charles River, and bred in-house at vivaria of Ghent University. Mice were housed in individually ventilated cages and kept under pathogen-free conditions. All animal experiments were conducted with permission of the Ethics Committee on Laboratory Animal Welfare of Ghent University.

## Reagents

The DPP4 inhibitor sitagliptin, the pan-DPP inhibitors (cyclic) VBP, the DPP8/DPP9 inhibitor 1G244, the FAP inhibitor UAMC1110 (Compound 60), the PREP inhibitor KYP-2047, and the DPP2 inhibitor UAMC39 (Compound 7.4) were synthesized at the Laboratory of Medicinal Chemistry of the University of Antwerp according to published protocols (Coutts et al, 1996; Jarho et al, 2004; Senten et al, 2004; Jiaang et al, 2005; Kim et al, 2005; Connolly et al, 2008; Jansen et al, 2014). Recombinant expression and purification of *B. anthracis* protective antigen (PA) was performed as described (von Moltke et al, 2012). *B. anthracis* lethal factor (LF) was acquired from List Biologicals (#172C). The antibodies used in

the study were against caspase-1 (AG-20B-0042-C100; Adipogen), caspase-8 (8592S, D5B2, and 4790; Cell Signaling), IL-1β (GTX74034; GeneTex), and ASC (AL177; Adipogen). HRP-conjugated anti-mouse (115-035-146) or anti–rabbit (111-035-144) antibodies were acquired from Jackson Immunoresearch Laboratories. Enhanced chemiluminescence solution, Sytox Green (S7020), anti-rabbit DyLight 594 (SA5-10040), and ProLong Gold Antifade Reagent with DAPI (P36935) were purchased from Thermo Fisher Scientific. LPS-SM (tlrl-smlps) was acquired from Invivogen. Bio-Plex kits Pro-Mouse IL-1β (171-G5002M) and Bio-Plex Pro-Mouse IL-18 (171-G6009M) were from Bio-Rad. Human DPP4 and DPP2 were purified from seminal plasma, as described (de Meester et al, 1996; Maes et al, 2005). A plasmid encoding human recombinant DPP9 (Dharmacon, Accession number: DQ892325) was expressed in Sf9 insect cells using the N-terminal BaculoDirect kit from Life Technologies. Recombinant DPP9 was purified using immobilized Ni-chelating chromatography (GE Healthcare), followed by anion-exchange chromatography using a 1 ml Mono Q (GE Healthcare). Human recombinant FAP was purchased from R&D (3715-SE). Human recombinant PREP was expressed in BL21(DE3) cells and purified using immobilized Co-chelating chromatography (GE

Healthcare) followed by anion-exchange chromatography on a 1 ml Mono Q column (GE Healthcare).

### Bone marrow–derived macrophage differentiation and stimulation

Macrophages were differentiated by culturing bone marrow progenitor cells in IMDM (Lonza) containing 10% (vol/vol) heat-inactivated FBS, 30% (vol/vol) L929 cell-conditioned medium, 1% (vol/vol) nonessential amino acids (Lonza), 100 U/ml penicillin and 100 mg/ml streptomycin at 37°C in a humidified atmosphere containing 5% $CO_2$ for 6 d. BMDMs were then seeded into 96- or 24-well plates as needed, in IMDM containing 10% FBS, 1% nonessential amino acids, and antibiotics. On the next day, cells were changed to fresh media and either primed or not with 100 ng/ml LPS for 3 h before treatment with vehicle or sitagliptin (water), 1G244 (DMSO), UAMC39 (water), UAMC1110 (DMSO), VBP (0.1%TFA in DMSO), acetyl VBP (DMSO), cyclic VBP (0.1%TFA in DMSO), or KYP-2047/UAMC714 (DMSO) at a final concentration of 10 $\mu$M unless otherwise stated in the figure legends. Alternatively, cells were stimulated with 1 $\mu$g/ml PA combined with 0.5 $\mu$g/ml LF (LeTx).

### J774.A1 culture and stimulation

J774.A1 cells were cultured in IMDM (Lonza) containing 10% (vol/vol) heat-inactivated FBS, 1% (vol/vol) nonessential amino acids (Lonza), 100 U/ml penicillin, and 100 mg/ml streptomycin at 37°C in a humidified atmosphere containing 5% $CO_2$. Cells were scraped for passages and, for experiments, counted and seeded into 96-well plates in culturing medium. On the next day, cells were changed to fresh media and stimulated with vehicle or 1 $\mu$g/ml PA combined with 0.5 $\mu$g/ml LF (LeTx), VBP (0.1% TFA in DMSO) or 1G244 (DMSO) at a final concentration of 10 $\mu$M or 25 $\mu$M.

### Analysis of inhibitor potency

Residual enzymatic activity in cell lysate and culture medium were measured as described previously (Matheeussen et al, 2012). Briefly, a calibration curve was created with known concentrations of the inhibitor diluted in the biological sample to be analyzed. The percentage of the residual enzymatic activity was calculated by comparing the enzymatic activity of inhibitor-treated cells with the activity of mock-treated cells (defined as 100% activity). To this end, 2 × 10[6] BMDMs were either mock-treated or stimulated with 10 $\mu$M of VBP, cVBP, sitagliptin, UAMC39, UAMC1110, KYP-2047, or 1G244 for 15 min and supernatants were collected. For cell lysates, BMDMs were washed once with PBS and scraped on lysis buffer (1% octylglucoside, 10 mM EDTA, 70 $\mu$g/ml aprotinin, and 50 mM Tris–HCl, pH 8.3). After 30 min, lysates were centrifuged at 12,000 *g* for 10 min at 4°C, and supernatants were collected for further analysis. Then, each enzyme (as described below) was incubated with its corresponding inhibitor or sample for 15 min at 37°C before the addition of preheated substrate solution. The residual enzymatic activity in either supernatant of cell lysates from BMDMs was calculated as the difference between a sample with exogenously added enzyme minus the sample without enzyme, thus correcting for background activity. In addition, the appropriate blanks were

included to correct for differences in background activity when constructing the calibration curves. Further experimental conditions are assay-dependent and are described below. Because all the inhibitors are reversible, the total dilution factor was considered. However, when preparing the cell lysates, an unknown dilution factor is introduced and the concentration calculated from the calibration curve will be inherently an estimation of the actual intracellular inhibitor concentration. All calibration curves were fitted using a nonlinear fit model in GraFit 7 software, according to the following equation:

$$y = \frac{range}{1 + \left(\frac{x}{IC_{50}}\right)^s} + background$$

where $y$ is the value of the residual activity compared with a non-inhibited sample, $x$ is the concentration of the inhibitor in the assay, $s$ is the slope factor, and $IC_{50}$ is the half maximal inhibitory concentration. The equation of the resulting graph was used to estimate the concentration of the inhibitor in the culture medium or in the cell lysate.

#### Sitagliptin, VBP, cVBP, and 1G244
DPP activity measurements were performed in buffer containing 0.1% Tween-20, 150 mM NaCl, 0.1 mg/ml BSA, 50 mM Tris–HCl, pH 8.3. Glycyl-prolyl-4-methoxy-$\beta$-naphthylamide (Gly-Pro-4Me$\beta$NA) was used as the substrate at a final concentration of 0.5 mM. For sitagliptin, the exogenous enzyme used was human DPP4; and for VBP, cVBP, and 1G244, human DPP9 was added. After the addition of substrate, the release of 4Me$\beta$NA was measured kinetically with $\lambda_{ex}$ = 340 nm and $\lambda_{em}$ = 430 nm wavelengths for 10 min at 37°C. The experiment was performed once and each sample/inhibitor was measured in duplicate.

#### UAMC39
DPP2 activity measurements were performed in buffer containing 0.1 M acetate (pH 5.5), 10 mM EDTA, and 14 $\mu$g/ml aprotinin. Lys-Ala-paranitroanilide (Lys-Ala-pNA) was used as substrate at a final concentration of 1 mM. After the addition of substrate, the release of pNA was measured kinetically with 405 nm wavelength for 10 min at 37°C. The experiment was performed twice, and each sample/inhibitor was measured in duplicate.

#### UAMC1110 and KYP-2047/UAMC714
FAP and PREP were assayed in buffer containing 0.1 M Tris–HCl (pH 8), 300 mM NaF, 1 mM EDTA, 50 mM salicylic acid, and 5 mM DTT. N-benzyloxycarbonyl-glycyl-prolyl-7-amido-4-methylcoumarin (Z-Gly-Pro-AMC) was used as the substrate at a final concentration of 266 $\mu$M. After the addition of substrate, the release of AMC was measured kinetically with $\lambda_{ex}$ = 380 nm and $\lambda_{em}$ = 465 nm wavelengths for 10 min at 37°C. The experiment was performed once, and each sample/inhibitor was measured in duplicate.

### Cell death kinetics

Analysis of cell death was performed through incorporation of 500 nM of Sytox Green dye in a 96-well format assay. Data were acquired

with a 20× objective using the Incucyte Zoom system (Essen Bio-Science) in a $CO_2$ and temperature-controlled environment. Each condition was run in technical duplicates. The number of fluorescent objects was counted with Incucyte ZOOM (Essen Bio-Science) software and plotted considering the highest value obtained in a well treated with Triton—X100 as 100%.

### Cytokine analysis

Cell culture supernatant was collected after 4, 8, and/or 24 h of stimulation, and culture medium was measured by magnetic bead–based multiplex assay using Luminex technology (Bio-Rad) according to the manufacturer's instructions. GraphPad Prism 6.0 software was used for data analysis.

### Western blotting

Cells lysed in Laemmli buffer on ice for 10 min. Supernatants were combined with cell lysates for detection of caspase-1 and IL-1$\beta$. For detection of caspase-8, most of the supernatant was removed. Subsequently, protein samples were boiled at 95°C for 10 min and separated by SDS–PAGE. Separated proteins were transferred to PVDF membranes. Blocking, incubation with antibody and washing of the membrane were performed in PBS supplemented with 0.05% or 0.1% Tween-20 (vol/vol) and 3% nonfat dry milk.

### ASC speck quantification and confocal imaging

BMDMs seeded on eight-well slide chambers (ibidi) were treated for 4 h or 8 h with VBP or 1G244 and fixed with 4% (wt/vol) paraformaldehyde. Cells were stained with anti-ASC (1:200) antibody, followed by secondary anti-rabbit DyLight 594 (1:1,000). Then, slides were mounted on ProLong Gold Antifade reagent with DAPI for analysis. For quantification of ASC specks, analyses were performed on a spinning-disk system (Zeiss), using an observer Z.1 microscope equipped with a Yokogawa disk CSU-X1. Fluorescent images were acquired with the use of a Plan-Apochromat 40×/1.4 oil DIC III objective and a Rolera Em-C2 Camera. Tile scanning was performed combining 4 × 5 images for the final mosaic (total area 7.42 mm × 5.91 mm). The maximum intensity projections of the final mosaic were analysed on Volocity 6.3.1 (PerkinElmer) software for the number of cells containing ASC specks. Seven mosaics were quantified per experimental condition from each biological repeat and the data from all repeats were combined. One-way ANOVA with Tukey's multiple comparisons analysis was performed in GraphPad Prism 6.0 software.

For bright-field imaging of cell death induction, BMDMs were seeded on eight-well slide chambers and treated for 4 h with VBP or 1G244. Imaging was performed on the same microscope as above, and cells were kept in a chamber with a 5% $CO_2$ atmosphere at 37°C throughout the experiment. Representative images were extracted and edited in Fiji (National Institutes of Health).

## Supplementary Information

## Acknowledgements

We thank Dr. Vishva M Dixit and Dr. Nobuhiko Kayagaki (Genentech), and Dr. Richard Flavell (Yale) for mutant mice; and the VIB Bioimaging Core Facility for technical support. This work was supported by FWO-SB grant no. 1S22417N to E De Hert, GOA BOF 2015 grant no. 30729 of the University of Antwerp to I De Meester and A-M Lambeir, DFG grant no GE2234/1-3 to R Geiss-Friedlander, and a European Research Council grant no. 683144 (PyroPop) and the Baillet Latour Medical Research Grant to M Lamkanfi.

## Author Contributions

NM de Vasconcelos: conceptualization, data curation, formal analysis, validation, investigation, visualization, methodology, and writing—original draft, review, and editing.
G Vliegen: formal analysis, validation, investigation, methodology, and writing—review and editing.
A Gonçalves: formal analysis, validation, investigation, methodology, and writing—review and editing.
E De Hert: formal analysis, validation, investigation, methodology, and writing—review and editing.
R Martín-Pérez: validation, investigation, methodology, and writing—review and editing.
N Van Opdenbosch: formal analysis, validation, investigation, methodology, and writing—review and editing.
A Jallapally: resources and writing—review and editing.
R Geiss-Friedlander: resources and writing—review and editing.
A-M Lambeir: resources and writing—review and editing.
K Augustyns: resources and writing—review and editing.
P Van Der Veken: resources and writing—review and editing.
I De Meester: conceptualization, resources, and writing—review and editing.
M Lamkanfi: conceptualization, resources, formal analysis, supervision, funding acquisition, validation, visualization, project administration, writing—original draft, review, and editing.

## Conflict of Interest Statement

N Van Opdenbosch, R Martín-Pérez, and M Lamkanfi are employees of Janssen Pharmaceutica. The authors declare that they have no conflict of interest.

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
