## [Reviewer comments · Life Science Alliance]

DPP8/DPP9 inhibition elicits canonical Nlrp1b inflammasome hallmarks in murine macrophages

Nathalia de Vasconcelos, Gwendolyn Vliegen, Amanda Gonçalves, Emilie De Hert, Rosa Martín-Pérez, Nina Van Opdenbosch, Anvesh Jallapally, Ruth Geiss-Friedlander, Anne-Marie Lambeir, Koen Augustyns, Pieter Van Der Veken, Ingrid De Meester, and Mohamed Lamkanfi

DOI: <https://doi.org/10.26508/lsa.201900313>

Corresponding author(s): Mohamed Lamkanfi, Janssen Pharmaceutica

Review Timeline:	Submission Date:	2019-01-21
	Editorial Decision:	2019-01-22
	Revision Received:	2019-01-24
	Editorial Decision:	2019-01-25
	Revision Received:	2019-01-25
	Accepted:	2019-01-25

Scientific Editor: Andrea Leibfried

Transaction Report:

Please note that the manuscript was previously reviewed at another journal and the reports were taken into account in inviting a revision for publication at *Life Science Alliance* prior to submission to *Life Science Alliance*.

Referee #1 Review

Received: 16th Dec 18

Report for Author:

In their manuscript 'DPP8/DPP9 inhibition elicits canonical inflammasome hallmarks in Nlrp1b-sufficient macrophages', de Vasconcelos et al. investigate Nlrp1 inflammasome assembly in response to treatment of murine macrophages with Dpp8/9 inhibitors.

The authors show that bone marrow-derived macrophages from C57BL/6J mice lose membrane integrity in response to treatment with inhibitors of Dpp8/9. This phenotype is enhanced in macrophages from a transgenic C57BL/6J mouse line that - in addition to their endogenous Nlrp1b (allele 2) - express Nlrp1b from 129S1 (from a BAC, i.e. its endogenous promoter). In macrophages with this genetic background, treatment with Dpp8/9 inhibitors triggered assembly of ASC specks, recruitment and processing of caspase-1, caspase-1- and GSDMD-dependent pyroptosis, and -if upregulated by LPS pretreatment- processing of pro-IL1 β and secretion of IL-1 β . The authors further show that macrophages expressing 129S1 Nlrp1b undergo apoptosis when caspase-1 or GsdmD are missing, although caspase-1-deficient cells are protected from cell death while GsdmD knockouts are not. The authors use well-controlled pharmacological experiments to prove that inflammasome assembly indeed is triggered by inhibition of Dpp8 and 9, and not any other potential targets of talabostat/Val-boroPro (VBP).

This study mostly confirms data previously shown for murine Nlrp1b in macrophage cell lines and macrophages of different genetic backgrounds, as well as for human NLRP1 in keratinocytes and mononuclear cells. While the new kinetic data for cell death and the dataset itself is interesting, the manuscript does not provide novel mechanistic insights, or technical approaches of broader interest, which would justify publication in this journal.

As the authors state, a number of recent papers revealed that endogenous NLRP1 in human keratinocytes and PBMCs, as well as ectopically expressed human NLRP1 in HEK 293T cells expressing ASC-EGFP, can assemble inflammasomes in response to Dpp8/9 inhibitors or dominant-negative mutants (Zhong et al., 2018). This resulted in the assembly of ASC specks, cytokine release, as well as induction of cell death by pyroptosis. Similarly, NLRP1b-dependent caspase-1 activation in murine C57BL/6J macrophages and the murine BALB/c macrophage cell line RAW264.7 (lacking ASC) have been shown to trigger cell death by pyroptosis (Okondo et al., 2017, 2018). Caspase-1 activation was unambiguously attributed to loss of Dpp8/9 activity (Okondo et al., 2017). While it was not previously shown that VBP triggers the assembly of ASC specks in murine macrophages, this was expected as NLRP1(b) activation in both species nucleates ASC specks: Both VBP-triggered activation of human NLRP1 (Zhong et al., 2016, 2018), and Lethal Factor-mediated activation of murine Nlrp1b yield ASC specks, activate caspase-1, and trigger pyroptosis and IL-1b secretion (Van Opendenbosch et al., 2014).

While it is possible that Nlrp1a and various Nlrp1b alleles respond differently to inhibitors of Dpp8/9, this is not addressed in this manuscript, as most experiments are performed in cells that express C57BL/6J NLRP1a and NLRP1b (allele 2), as well as NLRP1b (allele 1 from 129S1). Caspase-8 activation and apoptosis in response to inflammasome assembly in the

absence of caspase-1 or GSDMD has been described before (Schneider et al., 2017; Van Opendenbosch et al., 2017).

Taken together, the carefully assembled data in this paper does not add any novel mechanistic insights into the activation of NLRP1(b).

Other major points:

1. The authors describe the Nlrp1 allele of C57BL/6J as not functional, although the only available information is that it is not cleaved by B. anthracis Lethal Factor and that it leads to a weaker response to Toxoplasma gondii infection (Boyden and Dietrich, 2006; Ewald, Chavarria-Smith and Boothroyd, 2014). It is likely that a physiological activator of allele 2 of murine Nlrp1b has simply not been identified to date, and that different alleles have simply evolved to respond to different pathogens, as shown for rat NLRP1b (Cirelli et al., 2014). While it is an attractive hypothesis to assume that the inflammasome responses observed in C57BL/6J is governed by Nlrp1a, the authors do not perform any experiments that rule out a role of C57BL/6J Nlrp1b. This should be discussed more openly in the manuscript, avoiding strong conclusions on the functionality of Nlrp1b allele 2.

2. The authors perform one experiment with wt C57BL/6J and compare it to the C57BL/6J line expressing the 129S1 allele 1 of Nlrp1b. Due to the increased response to cyclic Val-boroPro and 1G244 (the response to Val-boroPro is rather similar), they conclude that experiments need to be done in the presence of a 'functional' copy of Nlrp1b. However, they cannot rule out that the enhanced response is simply explained by a dosage effect (Nlrp1b is in this case expressed from 3 genome copies) and should therefore tone down their claims. Nlrp1b signaling can be easily reconstituted in HEK 293T cells, and it would therefore be possible to compare the effect of Dpp8/9 inhibitors of all alleles and Nlrp1a.

3. Page 8, paragraph 2, lines 4-6: The authors claim: 'Notably, DPP inhibitors VBP and 1G244 were shown to induce pyroptosis without triggering the assembly of ASC specks in B6 macrophages and in the ASC deficient immortalized monocytic cell line RAW 264.72' by Okondo et al. (2017/2018). The authors of these papers simply did not test for ASC specks, which obviously does not rule out that they assemble. While pyroptosis may not require ASC, it is quite likely that ASC specks still occur in murine macrophages (see above). This needs to be corrected.

4. Almost all data points are described as 'mean {plus minus} SD of technical duplicates from three biological repeats'. While it is acceptable to only show one representative immunoblot, it should not be too difficult to represent the data from three independent ELISAs or to show at least some additional Sytox Green response curves as supplementary data. Standard deviations from two biological replicates are not particularly informative.

Minor points:

5. The authors state on page 4: 'whether DPP8/9 inhibition also promotes caspase-1 autocleavage, ASC speck formation, and secretion of mature IL-1 β and IL-18 from human mononuclear cells and keratinocytes is less clear'.

ASC specks (Fig. 2E), caspase-1 cleavage (2C), and cytokine secretion (2G) in human keratinocytes were clearly shown by Zhong et al. 2018.

6. After stimulation with Dpp8/9 inhibitors, Sytox Green was not able to stain nuclei of B6NLRP1b+ C1/11-/-, but did stain nuclei of B6NLRP1b+ GSDMD-/- macrophages, although both cells exhibit morphological features of apoptosis. The authors should discuss this. Due to the different functional outcomes, it would be helpful to test whether caspase-8 is also cleaved in the absence GsdmD. The authors should be more consistent in how they interpret SG staining: with regard to Fig 4b/c, they mention that apoptotic cells are not stained

with SG, while they interpret influx of SG as "late apoptosis" in the description of Fig 5b/c.

References

- Boyden, E. D. and Dietrich, W. F. (2006) 'Nalp1b controls mouse macrophage susceptibility to anthrax lethal toxin', *Nature Genetics*. Nature Publishing Group, 38(2), pp. 240-244. doi: 10.1038/ng1724.
- Cirelli, K. M. et al. (2014) 'Inflammasome Sensor NLRP1 Controls Rat Macrophage Susceptibility to *Toxoplasma gondii*', *PLoS Pathogens*. Edited by C. M. Sasseti. Public Library of Science, 10(3), p. e1003927. doi: 10.1371/journal.ppat.1003927.
- Ewald, S. E., Chavarria-Smith, J. and Boothroyd, J. C. (2014) 'NLRP1 is an inflammasome sensor for *Toxoplasma gondii*', *Infect Immun*, 82(1), pp. 460-468. doi: 10.1128/IAI.01170-13.
- Okondo, M. C. et al. (2017) 'DPP8 and DPP9 inhibition induces pro-caspase-1-dependent monocyte and macrophage pyroptosis', *Nature Chemical Biology*. Nature Publishing Group, 13(1), pp. 46-53. doi: 10.1038/nchembio.2229.
- Okondo, M. C. et al. (2018) 'Inhibition of Dpp8/9 Activates the Nlrp1b Inflammasome', *Cell Chemical Biology*. doi: 10.1016/j.chembiol.2017.12.013.
- Van Opdenbosch, N. et al. (2014) 'Activation of the NLRP1b inflammasome independently of ASC-mediated caspase-1 autoproteolysis and speck formation', *Nature Communications*. Nature Publishing Group, 5(1), p. 3209. doi: 10.1038/ncomms4209.
- Van Opdenbosch, N. et al. (2017) 'Caspase-1 Engagement and TLR-Induced c-FLIP Expression Suppress ASC/Caspase-8-Dependent Apoptosis by Inflammasome Sensors NLRP1b and NLRC4', *Cell Reports*. Cell Press, 21(12), pp. 3427-3444. doi: 10.1016/J.CELREP.2017.11.088.
- Schneider, K. S. et al. (2017) 'The Inflammasome Drives GSDMD-Independent Secondary Pyroptosis and IL-1 Release in the Absence of Caspase-1 Protease Activity', *Cell Reports*. Cell Press, 21(13), pp. 3846-3859. doi: 10.1016/J.CELREP.2017.12.018.
- Zhong, F. L. et al. (2016) 'Germline NLRP1 Mutations Cause Skin Inflammatory and Cancer Susceptibility Syndromes via Inflammasome Activation', *Cell*. Cell Press, 167(1), p. 187-202. doi: 10.1016/J.CELL.2016.09.001.
- Zhong, F. L. et al. (2018) 'Human DPP9 represses NLRP1 inflammasome and protects against autoinflammatory diseases via both peptidase activity and FIIND domain binding', *Journal of Biological Chemistry*, 293(49), pp. 18864-18878. doi: 10.1074/jbc.RA118.004350.

Referee #2 Review

Received: 3rd Jan 19

Report for Author:

The study "DPP8/DPP9 inhibition elicits canonical inflammasome hallmarks in Nlrp1b-sufficient macrophages" by Vasconcelos et al. provides evidence that pharmacological DPP8/9 inhibition elicits ASC-speck formation, caspase-1 maturation, and release of mature IL-1 β and IL-18 in murine macrophages. Previously published work reported that DPP8/9 inhibitors activate the Nlrp1 and CARD8 inflammasome. Since C57BL/6J mice lack a functional Nlrp1b allele, the authors here use their published mouse model in which Nlrp1b is reconstituted in C57BL/6J mice by hemizygous expression of a functional Nlrp1b allele from a 129S1-derived BAC. They find that Nlrp1b expression increases or accelerates the effects of inflammasome activation including ASC speck formation, but also find that, in contrast to *Bacillus anthracis* lethal toxin (Anthrax LT), DPP8/9 inhibitors still activate an inflammasome

even in the absence of Nlrp1b. Crossing Nlrp1b⁺ animals to mice deficient in downstream components of inflammasome biology (ASC, Caspase-1, GSDMD), they study the requirement of these players for the effects of DPP8/9 inhibitors. Using bone marrow-derived macrophages, the authors show that DPP8/DPP9 inhibition induces caspase-1 maturation and cytokine release dependent on the inflammasome adaptor protein ASC. They furthermore show that induction of pyroptosis is specific for DPP8/DPP9 inhibition, since specific inhibitors of other members of the dipeptidyl peptidase S9B family fail to induce pyroptosis and that DPP8/DPP9 inhibitory potency correlates with the ability to induce canonical inflammasome hallmarks. Lastly, the authors provide evidence that DPP8/DPP9 inhibition induces ASC-dependent apoptosis in caspase-1 deficient Nlrp1b sufficient macrophages that is initially also observed in Nlrp1b sufficient macrophages lacking Gasdermin D, but here, quickly proceeds to a lytic form of cell death not observed in the absence of caspase-1. The paper is generally well-written and the literature is discussed comprehensively. Experiments are largely conducted well, with the exception of the Western blots that show an extremely poor quality.

This study fills some experimental gaps left by previous work reporting the ability of DPP8/9 inhibitors to activate Nlrp1. However, the results are largely expected based on the general knowledge in the inflammasome field and the authors own previous work on Nlrp1b activation by Anthrax LT. The most interesting/novel/unexpected aspects of the story are the differences between Nlrp1b-proficient and deficient cells together with the differences between DPP8/9 inhibitors and Anthrax LT as well as the differences between caspase-1 and GSDMD deficiency in terms of the alternative cell death modalities engaged. Unfortunately, the authors choose not to follow up these aspects but rather, to remain within the confirmatory and expected. As such, the story is absolutely convincing, but not exiting. We recommend publication at a more specialized journal unless substantial novelty can be added, e.g. by investigating the role of Nlrp1a.

Major points:

1. The effect of DPP8/9 inhibitors and Anthrax LT on Nlrp1b proficient and deficient BMDMs should be systematically compared throughout the manuscript. At least for key data sets supporting central messages of the story, Anthrax LT and normal B6 cells (from wild-type and inflammasome knockout animals) should be included. For example, Figures 3C, D and S7 should also contain normal, unreconstituted bl/6 cells from wild-type and ASC^{-/-} mice and the supplement should be incorporated in the main figure.
2. The authors primarily use SG to measure pyroptosis, which allows kinetic measurements. This should be supplemented with LDH release assays throughout the manuscript for selected time points, which, again, should also include Anthrax LT.
3. The authors should include Anthrax LT in Figures 1 and 2 a-f in order to, on the one hand, confirm that hemizygous expression of a functional Nlrp1b allele from a 129S1-derived BAC in C57BL/6J mice also fully rescues Anthrax LT-induced pyroptosis under these conditions. On the other hand, the Anthrax LT positive control gives the reader the important opportunity to compare band intensities in the western blots and cytokine production levels to an established Nlrp1b activator.
4. Anthrax LT-induced inflammasome activation is not observed in bl/6 cells, while here, the authors show that also bl/6 cells respond to DPP8/9 inhibitors even without reconstitution of Nlrp1b. What is the molecular basis/receptor for that facilitates inflammasome formation/pyroptosis in bl/6 cells in the absence of Nlrp1b?
5. Fig. 2 and later figures: The Western blots don't distinguish between caspase-1 and IL-1b signal in the cell lysate and cell culture supernatant, as is standard in the inflammasome field. Cell lysate and supernatant samples should be harvested and analyzed separately for the key experiments within this manuscript. Furthermore, the Western blots seem inconsistent

between experiments with the ratio between Pro-caspase-1 and mature caspase-1 p20 changing between experiments. The manuscript tries to compare kinetics and degree of inflammasome activation in Nlrp1b-proficient and deficient cells and between different DPP inhibitors. The inconsistency of the Western blots suggests that this is not possible or at least does not reflect on the level of caspase-1 processing.

6. Fig 3: Although, formally, this was not shown since previous work used other experimental systems such as RAW cells that don't express ASC, the formation of specks is highly expected due to published data using Anthrax LT. Could it be that previously, ASC specks were not observed due to slower kinetics of speck formation in the absence of Nlrp1b in B6 cells? This should be investigated by live cell imaging of immortalized ASC-Cerulean reporter mouse BMDMs. What is the final frequency of specks in these cells induced by DPP8/9 inhibitors as compared to Anthrax LT and Nigericin or another soluble NLRP3 activator?

7. After having established in Fig 3a, b and before that DPP inhibitors activate an inflammasome in Nlrp1b-deficient cells, it is difficult to understand why the authors focus primarily on the investigation of Nlrp1b-reconstituted bl/6 mice (Fig 3f onward) instead of asking how DPP inhibitors (but not Anthrax LT) activate an inflammasome in normal bl/6 cells. This seems to be the much more interesting question. The authors discuss Nlrp1a as the likely factor involved, but don't perform any experiments to test this.

8. The authors only show DPP8/DPP9 inhibition specificity for induction of pyroptosis, but not for caspase-1 maturation and IL-1 β / IL-18 release. Showing that no caspase-1 cleavage to the active p20 fragment and cytokine release can be detected after treatment with specific inhibitors of other members of the dipeptidyl peptidase S9B family would help to support the conclusion that DPP8/DPP9 inhibition is causal to induction of canonical inflammasome hallmarks in Nlrp1b-sufficient macrophages.

9. Proteasome activity is required for Val-boroPro induced pyroptosis in RAW 264.7 cells. By pretreating the Nlrp1b sufficient macrophages with a proteasome inhibitor, it could be tested whether the effects shown in the present study also depend on proteasome activity.

10. The quality of the western blots shown is poor throughout the paper. Getting this under control and repeating experiments / rerunning the gels would contribute to a more convincing depiction of the results.

11. Fig 3: how does all this look in normal (non NLRP1b-expressing) B6 wildtype and ASC knockout mice? These experiments should contain Anthrax LT for direct comparison (if done in the same experiment, the data might be moved from the supplement for direct comparison).

12. Fig 3C, D: Why does ASC play a role for DPP8/9 but not for Anthrax LT? How does this look if LDH release is measured instead? How does this look without Nlrp1b?

13. Fig 3E, F: these blots are of extremely poor quality. As mentioned, these blots should be done either only from supernatant or both from supernatant and cell lysate samples (but not just from cell lysates) since the majority of mature caspase-1 leaves the cell during pyroptosis.

14. Fig 3 and last sentence in the results referring to this data on page 9: is caspase-1 required for DPP8/9 inhibitor-induced pyroptosis? Does this mean pro-caspase-1 is proteolytically active? GSDMD processing should be investigated in Nlrp1b-positive and negative ASC proficient and deficient cells. The data should be discussed in the context of the recent findings by Boucher and Schroder, JEM 2018.

15. Fig 4: how does the cell death modalities look in Nlrp1b negative caspase-1 wildtype or knockout cells?

16. Fig 4E, F: how does this look for pro-IL-1beta and/or IL-1alpha and/or HMGB1 (determined by ELISA or Western blot)?

17. Fig 5: The authors interpret cell lysis in GSDMD (but not Caspase-1) deficient cells as secondary necrosis. Necrosis is defined as a non-regulated form of cell death. Secondary necrosis is cell lysis after apoptosis in the absence of efferocytosis. However, efferocytosis is

absent for both genotypes here and the dependence of this lysis on Caspase-1 activity implies some kind of regulation. On this basis, the term secondary necrosis might not be full correct here.

18. Figure 4 and 5: is the same difference in cell death between GSDMD and Caspase-1 knockout cells also observed with Anthrax LT, or only with DPP-inhibitors?

19. Figure 4 and 5: can apoptosis (or even the lytic cell death observed in GSDMD knockouts) be blocked (or decelerated) by pan-caspase, Caspase-1 or Caspase-8 inhibitors?

20. Figure 4 and 5: the activation of Caspase-1, Caspase-8 and Caspase-3 should be systematically investigated in ASC, Caspase-1 and GSDMD-deficient Nlrp1b-positive (and ideally, negative) cells by Western blot.

21. Fig 5D, E: how does this look for pro-IL-1beta and/or IL-1alpha and/or HMGB1 (determined by ELISA or Western blot)?

Minor points:

1. Throughout the manuscript, the authors vary between the expressions "DPP8/DPP9 inhibition" and "DPP8/9 inhibition".

2. Page 3 ".which on its turn" - should this read "which in turn"?

3. Page 7, the authors call the caspase-1 cleavage observed "robust" while the signal observed in Fig 2 and elsewhere in the manuscript is rather weak and the Western blots of extremely poor quality.

4. Page 7, last sentence of first paragraph, an "of" is missing.

5. Page 8, text relating to Fig 3b, since the speck count in untreated cells ("mock") is basically zero, this is not an "increase" as the authors state, and anything is "significant" as compared to zero.

6. Please check if it is correct to refer to a cell as "intoxicated" (page 9).

7. The dye Sytox Green (SG) is intruded on page 9 of the main text in the context of Fig 4 but is already used in earlier figures and should be introduced at first appearance or not at all in the main text.

8. Page 9, last sentence: should this read "culture supernatants" instead of "cultured supernatants", since the cells, not the supernatants were cultured?

January 22, 2019

Re: Life Science Alliance manuscript #LSA-2019-00313-T

Prof. Mohamed Lamkanfi
VIB/Ghent University
Department of Medical Protein Research
Albert Baertsoenkaai 3
Ghent B-9000
Belgium

Dear Dr. Lamkanfi,

Thank you for transferring your manuscript entitled "DPP8/DPP9 inhibition elicits canonical inflammasome hallmarks in Nlrp1b-sufficient macrophages" to Life Science Alliance. The manuscript was assessed by expert reviewers at another journal before, and the editors transferred those reviewer reports to us with your permission.

The reviewers thought that while the kinetic data put forward are interesting, the broader conceptual advance provided by your findings remains limited. This is not a concern for publication in Life Science Alliance, and I would thus like to invite you to submit a revised version of your work. As outlined to you prior to submission to our journal, the concerns raised by reviewer #1 can get mainly addressed by changes to the text and by including an additional dataset on BALB/c-derived J774.A1 cells to exclude that the observations are due to dosage effects and to further assess the DPP8/9-associated inflammasome response in macrophages that uniquely express the LeTx-responsive 'functional' Nlrp1b allele 1. Furthermore, the term 'non-functional' should get avoided, please. The issues raised by reviewer #2 should get addressed as you already described in a preliminary point-by-point response, and the current discrepancy in figure 2 for the ratio of p45 and p20 caspase-1 bands after 8h should get addressed. Additionally, the following editorial points should get addressed:

- please mention the arrows of Fig 3A in the figure legend
- please mention the statistical test used in the legend for Fig 3b
- please add panel descriptors to figure legend S1
- please upload all S figures as individual files
- please list 10 authors et al in your reference list

When submitting the revision, please include a letter addressing the reviewers' comments point by

point.

Thank you for this interesting contribution to Life Science Alliance. We are looking forward to receiving your revised manuscript.

Sincerely,

- A letter addressing the reviewers' comments point by point.
- An editable version of the final text (.DOC or .DOCX) is needed for copyediting (no PDFs).
- High-resolution figure, supplementary figure and video files uploaded as individual files: See our detailed guidelines for preparing your production-ready images, <http://life-science-alliance.org/authorguide>
- Summary blurb (enter in submission system): A short text summarizing in a single sentence the study (max. 200 characters including spaces). This text is used in conjunction with the titles of papers, hence should be informative and complementary to the title and running title. It should describe the context and significance of the findings for a general readership; it should be written in the present tense and refer to the work in the third person. Author names should not be mentioned.

B. MANUSCRIPT ORGANIZATION AND FORMATTING:

Full guidelines are available on our Instructions for Authors page, <http://life-science-alliance.org/authorguide>

We sincerely thank the Editors and the Reviewer for providing helpful feedback to our preliminary point-by-point rebuttal letter. Please find a point-by-point response below that details how we have updated the manuscript in line with the comments and suggestions that we have received during the revision process. Most importantly, a new dataset of LeTx, VbP- and 1G244-induced pyroptosis in J774.A1 cells was added; the descriptor for Nlrp1b alleles was replaced with a more neutral/object term; and the Western blot in figure 2d was replaced by a more exposed version. We trust that with these changes you will find all outstanding remarks appropriately addressed, and hope the revised manuscript is now acceptable for publication.

Response to outstanding editorial and referee comments following our preliminary rebuttal letter:

(1) indeed include the BALB/c-derived J774.A1 dataset

The J774.A1 dataset is included as new Supplemental Fig. 5.

(2) choose another term than 'non-functional' when referring to Nlrp1b alleles 2, 3, and 4. The reviewer commented the following:

"I agree with most of the arguments of the authors, although I do not understand why the authors insist on the term 'non-functional' if other expressions could be used that do not bear the risk of being disproven at some point later. The term 'non-functional' suggest that NLRP1b of C57BL/J has some fundamental alterations that prevent it from being active (no expression, truncation, loss of crucial features such as the FIIND cleavage site), and thus implies that there is no need to test if NLRP1b allele 2 is activated by VBP. Yet, Chavarría-Smith et al. have shown that overexpression of NLRP1b allele 2 (C57BL/J) or N-terminal cleavage of an engineered TEV cleavage site in NLRP1b allele 2 leads to the recruitment and activation of caspase-1, which catalysed the maturation of pro-IL1 β (Chavarría-Smith and Vance, 2013).

By similar logic, human NLRP1 should also be called non-functional, since we do not know its physiological activator."

=> So I'd advice to choose another descriptor in this case.

We thank the Reviewer and Editor for the suggestion. We have replaced the term "functional" with the more neutral/objective term 'LeTx-responsive' throughout the manuscript.

(3) revise your figure 2 or address observed discrepancies in the text. The reviewer commented the following:

"If samples for both figures 2a-c and 2d were indeed generated the same way (as the figure legends implies), the ratio of p45 and p20 caspase-1 bands after 8h in 2a-c should be similar to the ratio of the same bands in 2d (same treatments and timing in both case). Do the bands look that different in all three independent replicates, or is it possible that figure 2d shows supernatants? This needs to be at least discussed."

We confirm that we have consistently used the same methodology to collect our samples for Western blots throughout the work (i.e. combined cell lysate+supernatant samples), also for the Western blots presented in figure 2. As discussed earlier, we think the differences the reviewer is commenting on reflect small differences in WB exposure times between independent experiments. We always capture different exposures of our Western blots, and

to accommodate the reviewer's concern, we have now substituted the originally presented Western blots in figure 2d for more exposed versions of the same blots, which more closely match the exposure times of the Western blots presented in panels a-c of figure 2. Although this change does not alter or impact in any way our conclusions drawn from these results, we hope incorporation of the more exposed Western blots in figure panel 2d satisfactorily addresses the comment that was raised.

Response to original editorial and referee comments:

Referee #1:

In their manuscript 'DPP8/DPP9 inhibition elicits canonical inflammasome hallmarks in Nlrp1b-sufficient macrophages', de Vasconcelos et al. investigate Nlrp1 inflammasome assembly in response to treatment of murine macrophages with Dpp8/9 inhibitors. The authors show that bone marrow-derived macrophages from C57BL/6J mice lose membrane integrity in response to treatment with inhibitors of Dpp8/9. This phenotype is enhanced in macrophages from a transgenic C57BL/6J mouse line that - in addition to their endogenous Nlrp1b (allele 2) - express Nlrp1b from 129S1 (from a BAC, i.e. its endogenous promoter). In macrophages with this genetic background, treatment with Dpp8/9 inhibitors triggered assembly of ASC specks, recruitment and processing of caspase-1, caspase-1- and GSDMD-dependent pyroptosis, and -if upregulated by LPS pretreatment- processing of pro-IL1 β and secretion of IL-1 β . The authors further show that macrophages expressing 129S1 Nlrp1b undergo apoptosis when caspase-1 or GsdmD are missing, although caspase-1-deficient cells are protected from cell death while GsdmD knockouts are not. The authors use well-controlled pharmacological experiments to prove that inflammasome assembly indeed is triggered by inhibition of Dpp8 and 9, and not any other potential targets of talabostat/Val-boroPro (VBP).

This study mostly confirms data previously shown for murine Nlrp1b in macrophage cell lines and macrophages of different genetic backgrounds, as well as for human NLRP1 in keratinocytes and mononuclear cells. While the new kinetic data for cell death and the dataset itself is interesting, the manuscript does not provide novel mechanistic insights, or technical approaches of broader interest, which would justify publication in this journal.

As the authors state, a number of recent papers revealed that endogenous NLRP1 in human keratinocytes and PBMCs, as well as ectopically expressed human NLRP1 in HEK 293T cells expressing ASC-EGFP, can assemble inflammasomes in response to Dpp8/9 inhibitors or dominant-negative mutants (Zhong et al., 2018). This resulted in the assembly of ASC specks, cytokine release, as well as induction of cell death by pyroptosis. Similarly, NLRP1b-dependent caspase-1 activation in murine C57BL/6J macrophages and the murine BALB/c macrophage cell line RAW264.7 (lacking ASC) have been shown to trigger cell death by pyroptosis (Okondo et al., 2017, 2018). Caspase-1 activation was unambiguously attributed to loss of Dpp8/9 activity (Okondo et al., 2017). While it was not previously shown that VBP triggers the assembly of ASC specks in murine macrophages, this was expected as NLRP1(b) activation in both species nucleates ASC specks: Both VBP-triggered activation of human NLRP1 (Zhong et al., 2016, 2018), and Lethal Factor-mediated activation of murine Nlrp1b yield ASC specks, activate caspase-1, and trigger pyroptosis and IL-1b secretion (Van Opdenbosch et al., 2014).

While it is possible that Nlrp1a and various Nlrp1b alleles respond differently to inhibitors of Dpp8/9, this is not addressed in this manuscript, as most experiments are performed in cells that express C57BL/6J NLRP1a and NLRP1b (allele 2), as well as NLRP1b (allele 1 from 129S1). Caspase-8 activation and apoptosis in response to inflammasome assembly in the absence of caspase-1 or GSDMD has been described before (Schneider et al., 2017; Van Opdenbosch et al., 2017).

Taken together, the carefully assembled data in this paper does not add any novel mechanistic insights into the activation of NLRP1(b).

We thank Referee #1 for the thorough analysis of our work and the clear summary of our findings. As pointed out by the Reviewer, we (Van Opdenbosch et al., 2014) have previously shown that LeTx-induced pyroptosis is associated with ASC speck formation, caspase-1 autocleavage and IL1 β /IL18 maturation and secretion in murine macrophages that express a LeTx-responsive 'functional' *Nlrp1b* allele (allele 1), whereas - contrastingly - macrophages that express a LeTx-unresponsive 'non-functional' *Nlrp1b* allele (allele 2 of C57BL/6 mice) fail to mount a LeTx-induced inflammasome response altogether (Van Opdenbosch et al., 2014). As also noted by the Referee, a recent study showed that DPP8/DPP9 inhibition also triggered NLRP1-dependent caspase-1 autocleavage, ASC speck formation and IL-1beta secretion in human keratinocytes (Zhong et al., 2018).

However, contrary to the inflammasome responses mentioned above, other recent work proposed that inhibition of DPP8/DPP9 in murine C57BL/6J macrophages (that express the LeTx-unresponsive 'non-functional' *Nlrp1b* allele 2), and in the murine BALB/c macrophage cell line RAW264.7 (that lacks expression of the central inflammasome adaptor ASC) triggered pyroptosis in the absence of other inflammasome hallmark responses, including caspase-1 autocleavage and IL-1beta secretion (Okondo et al., 2017, 2018). Given the physiological importance of IL-1beta maturation and secretion in inflammasome biology, we decided to systematically assess the effector mechanisms and the putative roles of ASC specks and caspase-1 autocleavage upon DPP8/DPP9 inhibition in murine macrophages that express the LeTx-responsive/' functional' *Nlrp1b* allele 1 in primary C57BL/6 macrophages (B6^{*Nlrp1b*+}). Our observations unequivocally demonstrate that DPP8/9 inhibition triggers NLRP1b-dependent pyroptosis that is associated with caspase-1 autocleavage, ASC speck formation and IL-1beta secretion. Thus, the presented work adds novel mechanistic insight by showing for the first time that all established inflammasome hallmarks (ASC speck assembly, caspase-1 maturation, IL1 β maturation and secretion) are engaged by DPP8/DPP9 inhibition in murine macrophages that express the LeTx-responsive 'functional' *Nlrp1b* allele 1. While we agree our results do not rule out a potential role for the 'LeTx-unresponsive 'non-functional' *Nlrp1b* allele 2 in the effects we have observed; they provide unequivocal genetic evidence that the LeTx-responsive 'functional' *Nlrp1b* allele is fully competent in driving all hallmark inflammasome responses in response to DPP8/9 inhibition.

To capture the above more clearly in the manuscript, we included the following statement in the introduction:

'A recent report showed that DPP8/DPP9 inhibition in human keratinocytes elicited the known hallmark features of canonical inflammasome activation, including caspase-1 autocleavage, ASC speck formation, and secretion of mature IL-1 β and IL-18 (Zhong et al, 2018). However, it is less clear whether these inflammasome responses are elicited upon DPP8/DPP9 inhibition in human and murine mononuclear cells (Johnson et al, 2018; Okondo et al, 2017; Zhong et al, 2018).'

If the Editors deem it necessary to further assess the DPP8/9-associated inflammasome response in macrophages that uniquely express the LeTx-responsive 'functional' *Nlrp1b* allele 1, we would be happy to include additional data sets from experiments in BALB/c-derived J774.A1 monocyte/macrophage cell line.

Other major points:

1. The authors describe the *Nlrp1* allele of C57BL/6J as not functional, although the only available information is that it is not cleaved by *B. anthracis* Lethal Factor and that it leads to a weaker response to *Toxoplasma gondii* infection (Boyden and Dietrich, 2006; Ewald, Chavarria-Smith and Boothroyd, 2014). It is likely that a physiological activator of allele 2 of murine *Nlrp1b* has simply not been

identified to date, and that different alleles have simply evolved to respond to different pathogens, as shown for rat NLRP1b (Cirelli et al., 2014). While it is an attractive hypothesis to assume that the inflammasome responses observed in C57BL/6J is governed by Nlrp1a, the authors do not perform any experiments that rule out a role of C57BL/6J Nlrp1b. This should be discussed more openly in the manuscript, avoiding strong conclusions on the functionality of Nlrp1b allele

We thank the reviewer for the comment. The *Nlrp1b* allele found in C57BL/6J mice (allele 2) is not only resistant to LeTx cleavage, but it also confers resistance to cell death induction, caspase-1 autocleavage, ASC speck formation and the induction of IL-1beta maturation and secretion by LeTx (Boyden & Dietrich, 2006; Van Opdenbosch et al., 2014).

Although we concur that it cannot be ruled out that the LeTx-unresponsive C57BL/6J-derived *Nlrp1b* allele 2 may have a yet undiscovered activity, no studies to date have formally established that this allele is capable of inducing inflammasome activation in response to physiological, pathogenic or pharmacological agents. To clarify this further and to define our use of the terms 'functional' and 'non-functional' in the manuscript, we inserted the following text to the introductory section:

“Although it cannot be ruled out that the LeTx-unresponsive C57BL/6J-derived Nlrp1b allele may have yet undiscovered activities, no studies to date have formally established that it is capable of eliciting inflammasome activation in response to endogenous, environmental, microbial and pharmacological agents, and inflammasome activation upon LeTx intoxication has only been formally demonstrated in the presence of allele 1 of Nlrp1b (Boyden & Dietrich, 2006; Van Opdenbosch et al, 2014).”

We also extended the discussion with the following statement:

“Nlrp1b allele 2 of C57BL/6J macrophages has not been decisively shown to support inflammasome assembly.”

2. The authors perform one experiment with wt C57BL/6J and compare it to the C57BL/6J line expressing the 129S1 allele 1 of Nlrp1b. Due to the increased response to cyclic Val-boroPro and 1G244 (the response to Val-boroPro is rather similar), they conclude that experiments need to be done in the presence of a 'functional' copy of Nlrp1b. However, they cannot rule out that the enhanced response is simply explained by a dosage effect (Nlrp1b is in this case expressed from 3 genome copies) and should therefore tone down their claims. Nlrp1b signaling can be easily reconstituted in HEK 293T cells, and it would therefore be possible to compare the effect of Dpp8/9 inhibitors of all alleles and Nlrp1a.

We respectfully disagree with the reviewer that the response to Val-boroPro is similar in wt C57BL/6J and C57BL/6J macrophages expressing the 129S1 allele 1 of Nlrp1b. Our results in Figure 1a clearly show that pyroptosis induction in wt C57BL/6J macrophages is significantly delayed when compared to cells that express the 129S1 allele 1 of Nlrp1b. It is evident in this regard that the time to reach a half-maximal induction of pyroptosis (35% cell death in the total cell population) is only about 3 hours for C57BL/6J macrophages expressing the 129S1 allele 1 of Nlrp1b, while it takes approximately 9h to reach the same level of cell death induction in wt C57BL/6J macrophages.

While we agree that the above result does not fully rule out a dosage effect, we have in the meantime established that DPP8/9 inhibition in the BALB/c-derived J774.A1 monocyte/macrophage cell line that has 2 genomic copies of the (functional) Nlrp1b allele 1 drives pyroptosis with similar kinetics to that of C57BL/6J macrophages expressing the 129S1 allele 1 of Nlrp1b. These results demonstrate that the functional 129S1 allele 1 of Nlrp1b likely accounted for the observed differential kinetic responses.

We would be happy to include the additional data set in BALB/c-derived J774.A1 monocyte/macrophage cell line or refer to it as data not shown if the Editors deem this necessary for publication.

3. Page 8, paragraph 2, lines 4-6: The authors claim: 'Notably, DPP inhibitors VBP and 1G244 were shown to induce pyroptosis without triggering the assembly of ASC specks in B6 macrophages and in the ASC deficient immortalized monocytic cell line RAW 264.72' by Okondo et al. (2017/2018). The authors of these papers simply did not test for ASC specks, which obviously does not rule out that they assemble. While pyroptosis may not require ASC, it is quite likely that ASC specks still occur in murine macrophages (see above). This needs to be corrected.

We have rephrased the statement to: "Notably, DPP inhibitors VBP and 1G244 were shown to induce pyroptosis in the ASC-deficient immortalized monocytic cell line RAW 264.7, and in ASC-deficient clones of the human monocytic cell line THP1 (Okondo et al, 2017). However, whether DPP8/DPP9 inhibition elicits ASC speck formation in ASC-sufficient macrophages and monocytes is not known."

4. Almost all data points are described as 'mean {plus minus} SD of technical duplicates from three biological repeats'. While it is acceptable to only show one representative immunoblot, it should not be too difficult to represent the data from three independent ELISAs or to show at least some additional Sytox Green response curves as supplementary data. Standard deviations from two biological replicates are not particularly informative.

Our data are representative of 3 independent experiments, and cytokine data are plotted as mean \pm SD of technical duplicates from three independent biological repeats. This is fully consistent to scientific standards, and a broadly adopted way of presenting these data types in the inflammasome field. To cite a random selection of recent reports in the high profile journals JEM and Nature Comms that used the same standard for presented data, see: Malliredi et al. J Exp Med. 2018;215(4):1023-1034; Lee et al. J Exp Med. 2018 Sep 3;215(9):2279-2288; Saavedra et al. Nat Commun. 2018:4846; Kanneganti et al. J Exp Med. 2018;215(6):1519-1529; Voet et al. Nat Commun. 2018 May 23;9(1):2036.

Minor points:

5. The authors state on page 4: 'whether DPP8/9 inhibition also promotes caspase-1 autocleavage, ASC speck formation, and secretion of mature IL-1 β and IL-18 from human mononuclear cells and keratinocytes is less clear'. ASC specks (Fig. 2E), caspase-1 cleavage (2C), and cytokine secretion (2G) in human keratinocytes were clearly shown by Zhong et al. 2018.

The reviewer correctly pointed out that the recent Zhong et al. 2018 paper showed that the cited hallmark features of inflammasome activation can be detected keratinocytes treated with DPP8/9 inhibitors. However, as discussed above, whether these hallmark features of inflammasome activation also occur in mononuclear cells (macrophages/monocytes) is currently being debated in the field given that Okondo et al. 2016 and Okondo et al. 2018 showed that caspase-1 automaturation does not occur following DPP8/9 inhibition in the human monocytic THP1 and U937 cell lines and in mouse C57BL/6J BMDMs. Furthermore, in the same publications, the authors claim IL-1 β is also not matured upon DPP8/9 inhibition in THP1 cells. ASC was demonstrated to be dispensable for cell death induction upon DPP8/9 inhibition, but whether ASC specks were formed was not addressed in this publication.

To make our statement clearer, we have rephrased the sentence in the manuscript as follows:

"A recent report showed that DPP8/DPP9 inhibition in human keratinocytes elicited the known hallmark features of canonical inflammasome activation, including caspase-1 autocleavage, ASC speck formation, and secretion of mature IL-1 β and IL-18 (Zhong et al, 2018). However, it is less clear whether these inflammasome responses are elicited upon DPP8/DPP9 inhibition in human and murine mononuclear cells (Johnson et al, 2018; Okondo et al, 2017; Zhong et al, 2018)."

6. After stimulation with Dpp8/9 inhibitors, Sytox Green was not able to stain nuclei of B6NLRP1b+ C1/11-/-, but did stain nuclei of B6NLRP1b+ GSDMD-/- macrophages, although both cells exhibit morphological features of apoptosis. The authors should discuss this. Due to the different functional outcomes, it would be helpful to test whether caspase-8 is also cleaved in the absence GsdmD. The

authors should be more consistent in how they interpret SG staining: with regard to Fig 4b/c, they mention that apoptotic cells are not stained with SG, while they interpret influx of SG as "late apoptosis" in the description of Fig 5b/c.

We thank the reviewer for the comment. We currently do not have an explanation for the differential effect between Casp1 and GsdmD KO cells. However, addressing whether Caspase-8 is activated in GSDMD^{-/-} BMDMs is beyond the scope of the current manuscript.

We have ensured to distinguish "early" and "late" apoptosis throughout the text, and added the following sentence to the discussion:

"Future work should investigate why DPP8/DPP9 inhibitor-induced apoptosis progresses significantly faster to a lytic phase (as marked by SG incorporation) in B6^{Nlrp1b+}GSDMD^{-/-} macrophages compared to apoptotic B6^{Nlrp1b+}C1^{-/-}C11^{-/-} BMDMs."

Referee #2:

The study "DPP8/DPP9 inhibition elicits canonical inflammasome hallmarks in Nlrp1b-sufficient macrophages" by Vasconcelos et al. provides evidence that pharmacological DPP8/9 inhibition elicits ASC-speck formation, caspase-1 maturation, and release of mature IL-1beta and IL-18 in murine macrophages. Previously published work reported that DPP8/9 inhibitors activate the Nlrp1 and CARD8 inflammasome. Since C57BL/6J mice lack a functional Nlrp1b allele, the authors here use their published mouse model in which Nlrp1b is reconstituted in C57BL/6J mice by hemizygous expression of a functional Nlrp1b allele from a 129S1-derived BAC. They find that Nlrp1b expression increases or accelerates the effects of inflammasome activation including ASC speck formation, but also find that, in contrast to Bacillus anthracis lethal toxin (Anthrax LT), DPP8/9 inhibitors still activate an inflammasome even in the absence of Nlrp1b. Crossing Nlrp1b⁺ animals to mice deficient in downstream components of inflammasome biology (ASC, Caspase-1, GSDMD), they study the requirement of these players for the effects of DPP8/9 inhibitors. Using bone marrow-derived macrophages, the authors show that DPP8/DPP9 inhibition induces caspase-1 maturation and cytokine release dependent on the inflammasome adaptor protein ASC. They furthermore show that induction of pyroptosis is specific for DPP8/DPP9 inhibition, since specific inhibitors of other members of the dipeptidyl peptidase S9B family fail to induce pyroptosis and that DPP8/DPP9 inhibitory potency correlates with the ability to induce canonical inflammasome hallmarks. Lastly, the authors provide evidence that DPP8/DPP9 inhibition induces ASC-dependent apoptosis in caspase-1 deficient Nlrp1b sufficient macrophages that is initially also observed in Nlrp1b sufficient macrophages lacking Gasdermin D, but here, quickly proceeds to a lytic form of cell death not observed in the absence of caspase-1. The paper is generally well-written and the literature is discussed comprehensively. Experiments are largely conducted well, with the exception of the Western blots that show an extremely poor quality. This study fills some experimental gaps left by previous work reporting the ability of DPP8/9 inhibitors to activate Nlrp1. However, the results are largely expected based on the general knowledge in the inflammasome field and the authors own previous work on Nlrp1b activation by Anthrax LT. The most interesting/novel/unexpected aspects of the story are the differences between Nlrp1b-proficient and deficient cells together with the differences between DPP8/9 inhibitors and Anthrax LT as well as the differences between caspase-1 and GSDMD deficiency in terms of the alternative cell death modalities engaged. Unfortunately, the authors choose not to follow up these aspects but rather, to remain within the confirmatory and expected. As such, the story is absolutely convincing, but not exciting. We recommend publication at a more specialized journal unless substantial novelty can be added, e.g. by investigating the role of Nlrp1a.

We thank Referee #2 for the thorough and clear summary of our work and evaluation of our contribution.

Major points:

1. The effect of DPP8/9 inhibitors and Anthrax LT on Nlrp1b proficient and deficient BMDMs should be systematically compared throughout the manuscript. At least for key data sets supporting central messages of the story, Anthrax LT and normal B6 cells (from wild-type and inflammasome knockout animals) should be included. For example, Figures 3C, D and S7 should also contain normal, unreconstituted bl/6 cells from wild-type and ASC^{-/-} mice and the supplement should be incorporated in the main figure.

Since we had previously reported a detailed analysis of the LeTx-induced inflammasome response in Nlrp1b proficient macrophages (Van Opdenbosch et al (2014) Nature Communications), we here focussed on characterizing the DPP8/9 inhibitor-induced inflammasome response in the same Nlrp1b proficient BMDM system. Moreover, we have compared the kinetics of DPP8/9 inhibition-induced pyroptosis in Nlrp1b proficient and C57BL/6J BMDMs in Figure 1. A comprehensive comparison in C57BL/6J cells throughout all of the different inflammasome hallmark features included in the manuscript was well beyond the scope of this work.

2. The authors primarily use SG to measure pyroptosis, which allows kinetic measurements. This should be supplemented with LDH release assays throughout the manuscript for selected time points, which, again, should also include Anthrax LT.

We thank the reviewer for the comment. SG staining is a more sensitive method to detect cell death compared to the LDH assay. We have previously compared both methods, and have shown that both SG incorporation and LDH release can be used as markers of pyroptotic cell death (de Vasconcelos et al., 2019). However, as the reviewer pointed out, SG allows for kinetic analyses of cell death in the same cell population, thus providing for a more informative method to track pyroptotic cell death, while this is less elegantly feasible with LDH assays. The SG staining analysis on the IncuCyte platform has been widely adopted by the cell death community to analyse lytic forms of cell death (both in the communities studying pyroptosis as the necroptosis research community) without generally requiring parallel LDH release assays. Thus, we do not believe an additional LDH analysis will enhance clarity of our data.

3. The authors should include Anthrax LT in Figures 1 and 2 a-f in order to, on the one hand, confirm that hemizygous expression of a functional Nlrp1b allele from a 129S1-derived BAC in C57BL/6J mice also fully rescues Anthrax LT-induced pyroptosis under these conditions. On the other hand, the Anthrax LT positive control gives the reader the important opportunity to compare band intensities in the western blots and cytokine production levels to an established Nlrp1b activator.

Please refer to our response to comment 1 of this referee.

4. Anthrax LT-induced inflammasome activation is not observed in bl/6 cells, while here, the authors show that also bl/6 cells respond to DPP8/9 inhibitors even without reconstitution of Nlrp1b. What is the molecular basis/receptor for that facilitates inflammasome formation/pyroptosis in bl/6 cells in the absence of Nlrp1b?

We thank the reviewer for raising this interesting topic. As discussed in the manuscript, we speculate that the functional Nlrp1a allele of C57BL/6J macrophages may account for the slow pyroptotic response that we have observed in wt C57BL/6J macrophages. Formal testing of this hypothesis awaits the generation of Nlrp1a deficient mice, which have not previously been reported, and we are currently generating. However, analysis of the response in this novel mouse strain is a project in its own right and well beyond the scope of the current manuscript.

5. Fig. 2 and later figures: The Western blots don't distinguish between caspase-1 and IL-1 β signal in the cell lysate and cell culture supernatant, as is standard in the inflammasome field. Cell lysate and supernatant samples should be harvested and analyzed separately for the key experiments within this manuscript. Furthermore, the Western blots seem inconsistent between experiments with the ratio between Pro-caspase-1 and mature caspase-1 p20 changing between experiments. The manuscript tries to compare kinetics and degree of inflammasome activation in Nlrp1b-proficient and deficient

cells and between different DPP inhibitors. The inconsistency of the Western blots suggests that this is not possible or at least does not reflect on the level of caspase-1 processing.

We thank the referee for the raised concerns. Throughout the whole manuscript, we have pooled cell lysates and supernatants to prepare the samples for WB analysis. This methodology aims at correcting for possible released proteins, as happens during pyroptosis, and has been extensively used in the field to interrogate inflammasome responses (as in Van Opdenbosch et al., 2017 (10.1016/j.celrep.2017.11.088), Kanneganti et al. 2018 (10.1084/jem.20172060), Malireddi et al., 2018 (10.1084/jem.20171922), Lee et al., 2018 (10.1084/jem.20180589)). To make this feature of our methodology clearer, we rephrased our Figure Legends that describe WB results to “Samples with combined cell lysate and supernatant were immunoblotted ...”. We also refer to our response to comment 4 of Reviewer 1.

We believe that what the reviewer has named as an inconsistency between the pro-caspase-1 and its p20 form throughout the manuscript actually reflects different experimental conditions of separate figures that have been performed as independent experiments: The WB conditions for each of the tested DPP8/9 inhibitors has been optimized in function of compound incubation time, concentration, and potency to induce inflammasome responses. We have used longer exposure times to develop WB for specific DPP inhibitors in order not to miss capturing potentially weaker levels of caspase-1 cleavage. Accordingly, our customized Western blotting conditions capture mature caspase-1 species (Fig 2) in all conditions in which we have also observed pyroptosis induction by independent methodologies (Fig 1). We further have established genetically that caspase-1 is indeed responsible for the cytotoxicity observed after DPP8/DPP9 inhibition in B6^{Nlrp1b+} macrophages (Fig 4). The same logic was used to optimize the WB conditions for the blots presented in Figure 2d (with lower WB exposure times) and in Figure 3e and 3f (with longer exposure times).

6. Fig 3: Although, formally, this was not shown since previous work used other experimental systems such as RAW cells that don't express ASC, the formation of specks is highly expected due to published data using Anthrax LT. Could it be that previously, ASC specks were not observed due to slower kinetics of speck formation in the absence of Nlrp1b in B6 cells? This should be investigated by live cell imaging of immortalized ASC-Cerulean reporter mouse BMDMs. What is the final frequency of specks in these cells induced by DPP8/9 inhibitors as compared to Anthrax LT and Nigericin or another soluble NLRP3 activator?

We thank the reviewer for the comment. We refer to our response to the related first comment of Reviewer 1 for a more extensive discussion. In brief: as noted above and in the manuscript, we and others have previously demonstrated that LeTx triggers assembly of ASC specks in Nlrp1b proficient but not in wildtype C57BL/6 macrophages. The current work focussed on the characterizing ASC speck formation by DPP8/9 inhibition in Nlrp1 proficient cells given that this has not been reported to occur in myeloid cells and given that the induction of other hallmark features such as caspase-1 automaturation and IL-1beta secretion by DPP8/9 inhibition is currently debated in the field in the context of the claims of the Okondo et al 2016 and 2018 reports.

7. After having established in Fig 3a, b and before that DPP inhibitors activate an inflammasome in Nlrp1b-deficient cells, it is difficult to understand why the authors focus primarily on the investigation of Nlrp1b-reconstituted bl/6 mice (Fig 3f onward) instead of asking how DPP inhibitors (but not Anthrax LT) activate an inflammasome in normal bl/6 cells. This seems to be the much more interesting question. The authors discuss Nlrp1a as the likely factor involved, but don't perform any experiments to test this.

We thank the reviewer for this interesting remark. We concur that it would be of interest to address which inflammasome is responsible for the observed cell death elicited by DPP8/9 inhibition in C57BL/6 cells. As hypothesized in the discussion section of the manuscript, we think this may be an Nlrp1a driven response. However, formal testing of this hypothesis awaits generation of Nlrp1a

deficient mice, which we are currently pursuing. In the context of the current work, we believed that it would be of interest to the inflammasome and cell death research communities to provide firm evidence whether (1) DPP8/9 pyroptosis was induced by unprocessed pro-caspase-1 or is accompanied by caspase-1 autocleavage, and (2) whether DPP8/9 inhibition elicits IL1 β /IL18 maturation and secretion; and (3) ASC speck formation in Nlrp1b proficient macrophages. These were essential gaps in the understanding of DPP8/9-initiated inflammasome response that have been conclusively addressed by the presented work.

8. The authors only show DPP8/DPP9 inhibition specificity for induction of pyroptosis, but not for caspase-1 maturation and IL-1 β / IL-18 release. Showing that no caspase-1 cleavage to the active p20 fragment and cytokine release can be detected after treatment with specific inhibitors of other members of the dipeptidyl peptidase S9B family would help to support the conclusion that DPP8/DPP9 inhibition is causal to induction of canonical inflammasome hallmarks in Nlrp1b-sufficient macrophages.

We thank the reviewer for the comment. In Figures 3, 4 and 5, we have provided conclusive genetic evidence in macrophages from inflammasome gene knockout mice to unequivocally demonstrate the role of relevant inflammasome components (i.e. ASC, Caspase-1/-11 and GSDMD) in the pyroptosis, caspase-1 autocleavage, ASC speck assembly and cytokine release in response to DPP8/9 inhibition. We believe these data sets decisively establish the link between DPP8/9 inhibition and the documented inflammasome responses.

9. Proteasome activity is required for Val-boroPro induced pyroptosis in RAW 264.7 cells. By pretreating the Nlrp1b sufficient macrophages with a proteasome inhibitor, it could be tested whether the effects shown in the present study also depend on proteasome activity.

It is currently unclear how proteasome inhibition regulates both LeTx-induced and DPP8/9 inhibition induced Nlrp1b inflammasome activation. Characterization of the proteasome and related mechanisms that act upstream of Nlrp1b is beyond the scope of the current study.

10. The quality of the western blots shown is poor throughout the paper. Getting this under control and repeating experiments / rerunning the gels would contribute to a more convincing depiction of the results.

We refer to our response to comment 5 of this referee.

11. Fig 3: how does all this look in normal (non NLRP1b-expressing) B6 wildtype and ASC knockout mice? These experiments should contain Anthrax LT for direct comparison (if done in the same experiment, the data might be moved from the supplement for direct comparison.

We refer to our response to comment 1 of this referee.

12. Fig 3C, D: Why does ASC play a role for DPP8/9 but not for Anthrax LT? How does this look if LDH release is measured instead? How does this look without Nlrp1b?

We thank the reviewer for the comment. We currently do not know why ASC plays a more important role during DPP8/9 inhibition-induced pyroptosis compared to the pyroptosis response induced by Anthrax LeTx. A possibility is that such responses might be linked to differential upstream pathways induced by these agents. Future research should address this and other possibilities.

13. Fig 3E, F: these blots are of extremely poor quality. As mentioned, these blots should be done either only from supernatant or both from supernatant and cell lysate samples (but not just from cell lysates) since the majority of mature caspase-1 leaves the cell during pyroptosis.

We thank the reviewer for carefully analysing our WBs. We apologize for not having pointed out more clearly that our WB samples have been prepared by combining cell lysate and supernatants, as is routinely done in the inflammasome field (e.g. Van Opdenbosch et al., 2017 (10.1016/j.celrep.2017.11.088), Kanneganti et al. 2018 (10.1084/jem.20172060), Malireddi et al., 2018 (10.1084/jem.20171922), Lee et al., 2018 (10.1084/jem.20180589)). We have added the

following sentence describing our methods to relevant figure legends: “Samples with combined cell lysate and supernatant were immunoblotted ...”. We also refer to our response to comment 5 of this Reviewer.

14. Fig 3 and last sentence in the results referring to this data on page 9: is caspase-1 required for DPP8/9 inhibitor-induced pyroptosis? Does this mean pro-caspase-1 is proteolytically active? GSDMD processing should be investigated in Nlrp1b-positive and negative ASC proficient and deficient cells. The data should be discussed in the context of the recent findings by Boucher and Schroder, JEM 2018.

We thank the reviewer for these questions. We and others have previously shown that genetic deletion of ASC prevents LeTx-induced pro-caspase-1 maturation in NLRP1b proficient cells; and that unprocessed pro-caspase-1 retains activity to induce pyroptosis under these conditions, similar to pro-caspase-1 activity by the NLRC4 inflammasome in ASC-deleted macrophages (Broz et al., 2010 (doi: 10.1016/j.chom.2010.11.007); Van Opdenbosch et al., 2014 (doi: 10.1038/ncomms4209); Guey et al., 2014 (doi: 10.1073/pnas.1415756111)). We have not addressed directly the activity of caspase-1 in our samples. As Boucher and Schroder, 2018 show, presence of caspase-1 p20 form in lysates is the result of (past) caspase-1 activity. Furthermore, we established caspase-1 is the pyroptotic mediator in B6^{Nlrp1+} BMDMs upon DPP8/9 inhibition by genetic deletion of this caspase, as shown in Figure 4.

15. Fig 4: how does the cell death modalities look in Nlrp1b negative caspase-1 wildtype or knockout cells?

We appreciate the reviewer’s interest. However, addressing the inflammasome responsible for pyroptosis upon DPP8/9 inhibition in the absence of a functional NLRP1b is beyond the scope of the current manuscript. Please also refer to our response to major comment 1 of this referee.

16. Fig 4E, F: how does this look for pro-IL-1beta and/or IL-1alpha and/or HMGB1 (determined by ELISA or Western blot)?

While interesting, addressing alarmin release upon DPP8/DPP9 inhibition was beyond the scope of the current study.

17. Fig 5: The authors interpret cell lysis in GSDMD (but not Caspase-1) deficient cells as secondary necrosis. Necrosis is defined as a non-regulated form of cell death. Secondary necrosis is cell lysis after apoptosis in the absence of efferocytosis. However, efferocytosis is absent for both genotypes here and the dependence of this lysis on Caspase-1 activity implies some kind of regulation. On this basis, the term secondary necrosis might not be full correct here.

We appreciate the reviewer’s point. Our designation of the observed effect as secondary necrosis is based on how literature uses this nomenclature, as in cell permeabilization of late apoptotic cells. Although we don’t have a clear explanation for the observed differences in Casp1/11 Ko and GSDMD KO cells, we agree that the difference observed between the two genotypes suggest that plasma membrane permeabilization after DPP8/9 inhibition-induced apoptosis is somehow controlled. Future research should address this. Please also refer to our response to related minor comment 6 of Referee 1.

18. Figure 4 and 5: is the same difference in cell death between GSDMD and Caspase-1 knockout cells also observed with Anthrax LT, or only with DPP-inhibitors?

We thank the reviewer for the question. Indeed, we see a similar effect (i.e. rapid plasma membrane permeabilization after apoptosis) happening upon activation of NLRP1b by LeTx and NLRC4. However, these data are associated to a different study that is being prepared for separate publication, and further inquiries on the mechanism that might be behind such differences is beyond the scope of the paper.

19. Figure 4 and 5: can apoptosis (or even the lytic cell death observed in GSDMD knockouts) be blocked (or decelerated) by pan-caspase, Caspase-1 or Caspase-8 inhibitors?

Please refer to our response to comment 18 of this referee.

20. Figure 4 and 5: the activation of Caspase-1, Caspase-8 and Caspase-3 should be systematically investigated in ASC, Caspase-1 and GSDMD-deficient Nlrp1b-positive (and ideally, negative) cells by Western blot.

As our goal was to characterize hallmark inflammasome functions upon DPP8/9 inhibition in B6^{Nlrp1b+} BMDMs, we did not consider it essential to include a comprehensive analysis of caspase cleavage in our panels beyond caspase-1.

21. Fig 5D, E: how does this look for pro-IL-1beta and/or IL-1alpha and/or HMGB1 (determined by ELISA or Western blot)?

As indicated in our response to comment 16 of this referee, addressing alarmin release upon DPP8/DPP9 inhibition was beyond the scope of the current study.

Minor points:

1. Throughout the manuscript, the authors vary between the expressions "DPP8/DPP9 inhibition" and "DPP8/9 inhibition".

We have adjusted all references to DPP8/DPP9.

2. Page 3 ".which on its turn" - should this read "which in turn"?

This has been corrected.

3. Page 7, the authors call the caspase-1 cleavage observed "robust" while the signal observed in Fig 2 and elsewhere in the manuscript is rather weak and the Western blots of extremely poor quality.

Adjusted or deleted.

4. Page 7, last sentence of first paragraph, an "of" is missing.

Adjusted.

5. Page 8, text relating to Fig 3b, since the speck count in untreated cells ("mock") is basically zero, this is not an "increase" as the authors state, and anything is "significant" as compared to zero.

Thank you for the remark, it has not been deleted.

6. Please check if it is correct to refer to a cell as "intoxicated" (page 9).

7. The dye Sytox Green (SG) is intruded on page 9 of the main text in the context of Fig 4 but is already used in earlier figures and should be introduced at first appearance or not at all in the main text.

Corrected.

8. Page 9, last sentence: should this read "culture supernatants" instead of "cultured supernatants", since the cells, not the supernatants were cultured?

Corrected.

January 25, 2019

RE: Life Science Alliance Manuscript #LSA-2019-00313-TR

Prof. Mohamed Lamkanfi
Janssen Pharmaceutica
Turnhoutseweg 30
Beerse 2340
Belgium

Dear Dr. Lamkanfi,

Thank you for submitting your revised manuscript entitled "DPP8/DPP9 inhibition elicits canonical Nlrp1b inflammasome hallmarks in murine macrophages". I appreciate the introduced changes and would thus be happy to publish your paper in Life Science Alliance. Before sending you the official acceptance letter, please log into our system one more time to fill in the electronic license to publish form. Your manuscript number will change to LSA-2019-00313-TRR, please make sure that all manuscript files get carried over to this new manuscript number (single click).

Please log in to your account: <https://lsa.msubmit.net/cgi-bin/main.plex>

A. FINAL FILES:

-- High-resolution figure, supplementary figure and video files uploaded as individual files: See our detailed guidelines for preparing your production-ready images, <http://life-science-alliance.org/authorguide>

B. MANUSCRIPT ORGANIZATION AND FORMATTING:

Full guidelines are available on our Instructions for Authors page, <http://life-science-alliance.org/authorguide>

Sincerely,

Andrea Leibfried, PhD
Executive Editor
Life Science Alliance
Meyershofstr. 1
69117 Heidelberg, Germany
t +49 6221 8891 502
e a.leibfried@life-science-alliance.org
www.life-science-alliance.org

January 25, 2019

RE: Life Science Alliance Manuscript #LSA-2019-00313-TRR

Prof. Mohamed Lamkanfi
Janssen Pharmaceutica
Turnhoutseweg 30
Beerse 2340
Belgium

Dear Dr. Lamkanfi,

Thank you for submitting your Research Article entitled "DPP8/DPP9 inhibition elicits canonical Nlrp1b inflammasome hallmarks in murine macrophages". It is a pleasure to let you know that your manuscript is now accepted for publication in Life Science Alliance. Congratulations on this interesting work.

*****IMPORTANT:** If you will be unreachable at any time, please provide us with the email address of an alternate author. Failure to respond to routine queries may lead to unavoidable delays in publication.*******

DISTRIBUTION OF MATERIALS:

Again, congratulations on a very nice paper. I hope you found the review process to be constructive and are pleased with how the manuscript was handled editorially. We look forward to future exciting submissions from your lab.

Sincerely,
